# The integrated stress response suppresses PINK1-dependent mitophagy by preserving mitochondrial import efficiency

Mingchong Yang [1,5], Zengshuo Mo[1,5], Kelly Walsh[1], Wen Liu[2], Imane Nait Irahal[3,4], Damien Arnoult [3] & Xiaoyan Guo [1] ✉

Mitophagy is crucial for maintaining mitochondrial health, but how its levels adjust to different stress conditions remains unclear. In this study, we investigated the role of the DELE1-HRI axis of the integrated stress response (ISR) in regulating mitophagy, a key mitochondrial quality control mechanism. Our findings show that the ISR suppresses PINK1-dependent mitophagy under many mitochondrial stress conditions by maintaining mitochondrial presequence protein import, independent of ATF4 activation. Mitochondrial presequence protein import efficiency is tightly linked to the rate of protein synthesis. Without the ISR, increased protein synthesis overwhelms the mitochondrial import machineries, reducing import efficiency. This impairment can be mitigated by pharmacological attenuation of protein synthesis, such as with mTOR or general translation inhibitors. Under severe depolarizing stress, mitochondrial import is heavily impaired even with an active ISR, leading to significant PINK1 accumulation. In contrast, mild mitochondrial stress allows more efficient protein import in the presence of the ISR, resulting in lower mitophagy. Without the ISR, mitochondrial protein import becomes significantly compromised, causing PINK1 accumulation to reach the threshold level necessary to trigger mitophagy. These findings reveal a link between ISR-regulated protein synthesis, mitochondrial protein import, and mitophagy, offering potential therapeutic targets for diseases associated with mitochondrial dysfunction.

Mitochondria, best known as the powerhouse of the cell, play crucial roles beyond energy production, including metabolic regulation, calcium homeostasis, and apoptosis[1]. To maintain mitochondrial function and cellular homeostasis, mitochondria engage in extensive crosstalk with other cellular components[2]. One key molecular pathway activated in response to most of mitochondrial stress conditions, is the integrated stress response (ISR)[3,4]. The ISR is a general adaptive mechanism that cells use to cope with a broad range of stress conditions, including viral infections, amino acid deprivation, heme depletion, and endoplasmic reticulum stress[5]. While ISR pathways are triggered by distinct stimuli, they all converge at a common regulatory node: the phosphorylation of eIF2α by one of four stress-specific kinases (HRI, PERK, GCN2, or PKR), resulting in global attenuation of protein synthesis while selectively upregulating stress-associated transcription factors, such as ATF4[6–8]. In the context of mitochondrial dysfunction, the key molecule to trigger the ISR is DELE1[9–11]. DELE1 localization

[1]Department of Genetics and Genome Sciences, University of Connecticut Health Center, Farmington, CT, USA. [2]Department of Molecular Biology and Biophysics, University of Connecticut Health Center, Farmington, CT, USA. [3]INSERM UMR-S-MD 1197, Université Paris Saclay, Villejuif, France. [4]Laboratoire Santé, Environnement et Biotechnologie, Faculté Des Sciences Ain Chock, Université Hassan II de Casablanca, BP5366 Maarif, Casablanca, Marocco. [5]These authors contributed equally: Mingchong Yang, Zengshuo Mo. ✉e-mail: xguo@uchc.edu

serves as a signal for sensing and relaying mitochondrial stress. Under conditions such as mitochondrial depolarization (e.g., CCCP) or ATP synthase inhibition (e.g., oligomycin), DELE1 during its import is cleaved by a mitochondrial inner membrane protease, OMA1. The cleaved product of DELE1 accumulates in the cytosol, where it oligo-merizes to form a scaffold platform that recruits and activates the eIF2α kinase HRI[12]. Under iron deficiency conditions, the full-length DELE1 is stabilized on the outer mitochondrial membrane (OMM) to activate HRI[11]. While the molecular mechanisms underlying ISR activation triggered by mitochondrial dysfunction are well understood, the role of DELE1-mediated ISR activation in regulating mitochondrial homeostasis remains unclear.

Mitophagy, a specialized form of autophagy that selectively eliminates damaged mitochondria, is a critical mechanism of mito-chondrial quality control[13,14]. Deficient mitophagy, leading to the accumulation of dysfunctional mitochondria, is linked to diverse human disorders, including neurodegenerative diseases such as Parkinson's Disease (PD)[15]. This link is highlighted by mutations in two genes PINK1 and PRKN, which cause familial PD[16,17]. PINK1 and PRKN play central roles in mitophagy[18]. PINK1 encodes for PTEN-induced putative kinase 1 (PINK1), and PRKN encodes a E3-ubiquitin ligase, PRKN. Under normal conditions, PINK1 is efficiently imported into healthy mitochondria. During import, PINK1 is cleaved by the matrix processing peptidase (MPP), and PARL, a mitochondrial pro-tease located in the inner membrane[19–21]. The cleaved form of PINK1 is subsequently released into the cytoplasm and degraded by the pro-teasome following the N-end rule[22]. However, when PINK1 import is disrupted due to mitochondrial damage, it can stabilize on the OMM, where it phosphorylates ubiquitin and PRKN[23,24]. As a result, PRKN is activated and tethered on the OMM, where it further ubiquitinates OMM proteins[25–27], marking them for degradation by proteasomes and for recognition by the autophagy receptors to initialize autop-hagosome formation. This ultimately leads to the engulfment and degradation of the damaged mitochondria[28–30]. The PINK1-PRKN-dependent mitophagy pathway is robustly triggered by severe mitochondrial depolarization, but is less responsive to other rela-tively mild stress conditions, such as oligomycin, rotenone or anti-mycin A treatment[24,31].

Here, we investigated the relationship between the DELE1-HRI axis of the ISR and PINK1-PRKN-dependent mitophagy under differ-ent stress conditions. Strikingly, we found that loss of the ISR com-ponents, such as OMA1, DELE1, HRI, and eIF2α phosphorylation, but not ATF4, robustly activates PINK1-dependent mitophagy under mild stress conditions through PINK1 stabilization. ISRIB, a small molecule inhibitor of the ISR that reverses the effects of eIF2α phosphorylation[32,33], can also enhance mitophagy, albeit to a lesser extent compared to genetic ablation. We further showed that PINK1 stabilization is coupled with impaired mitochondrial pre-sequence protein import in the absence of ISR, but without a strong effect on mitochondrial membrane potential. We speculate that attenuation of general protein synthesis due to the ISR activation reduces the protein influx into mitochondria, resulting in efficient mitochondrial import, even in the presence of mild mitochondrial stress. Consequently, mitophagy is inhibited under mild stress conditions because of efficient PINK1 import and destabilization. Indeed, mildly reducing protein synthesis in ISR-deficient cells can improve mitochondrial presequence protein import, decrease PINK1 stability and suppress mitophagy. Similarly, triggering alternative ISR pathways can suppress the mitophagy phenotype in cells without DELE1 axis of the ISR, sug-gesting that the ISR pathways in general can regulate mitochondrial protein import and mitophagy. Lastly, both the quality and quantity of mitochondria must be tightly regulated, and overactivation of mitophagy in the absence of ISR can be detrimental under certain stress conditions.

## Results

### DELE1 negatively regulates mitophagy

To monitor mitophagy, we stably expressed mtKeima in a HEK293T cell line equipped with CRISPRi machinery (Fig. 1a). mtKeima encodes a mitochondrially localized red fluorescent protein with dual excitation wavelengths that vary depending on the pH of its environment. It can be excited at 440 nm, a peak predominant at pH above 6 as found in mitochondria, and at 586 nm, a peak predominant at pH below 5 as in lysosomes[34]. We measured and quantified mitophagy using flow cytometry and defined a mitophagy cell population with higher mtKeima (lysosomes)/mtKeima (mitochondria) ratio (Fig. 1a). CRISPRi machinery allows for the knockdown of candidate genes via expres-sing gene-specific sgRNAs[35]. In HEK293T cells with a non-targeting control sgRNA (NTC), which serves as a wild type (WT) control, we observed a very low basal level of mitophagy (about 2%). Oligomycin, an ATP synthase inhibitor, or carbonyl cyanide m-chlorophenyl hydrazone (CCCP), a mitochondrial uncoupler that depolarizes mito-chondrial membrane potential[36], only induce minimal mitophagy (Supplementary Fig. 1a). Interestingly, knockdown of DELE1 (DELE1 KD) enhances basal levels of mitophagy (from 2% to 10%), which is further amplified slightly by oligomycin treatment (about 15%), but not by CCCP treatment (Supplementary Fig. 1a), suggesting that DELE1 may suppress mitophagy under mild stress conditions such as oligomycin treatment.

We speculate that low levels of mitophagy observed in HEK293T may result from weak endogenous expression of PRKN. Next, we tested whether overexpression (OE) of PRKN would lead to a stronger phe-notype. We lentivirally integrated miRFP-PRKN into the mtKeima cell lines and established three monoclonal cell lines with PRKN expression at low (PRKN OE^LL), medium (PRKN OE^ML), and high (PRKN OE^HL) levels, respectively (Fig. 1b, and Supplementary Fig. 1b). We also generated another PRKN OE cell line via integration of miRFP-PRKN into the AAVS-1 safe harbor locus (PRKN OE^AAVS1-HL), the expression levels and phe-notypes of this cell line are comparable to the monoclonal PRKN OE^HL (Supplementary Fig. 1b, 1e and 1f). We measured mitophagy in these cells across a range of oligomycin and CCCP concentrations. In both WT and DELE1 KD cells, oligomycin and CCCP treatment promote mitophagy in a manner dependent on drug concentrations and PRKN expression levels (Fig. 1c–f, and Supplementary Fig. 1c–h). Strikingly, mitophagy levels are consistently higher in DELE1 KD than WT cells across different oligomycin concentrations, regardless of PRKN expression levels, although this effect was more pronounced in cells with higher levels of PRKN (Fig. 1c–f). While both WT and DELE1 KD cells show similar levels of mitophagy following treatment with 10 μM CCCP (Fig. 1d–f), or co-treatment with oligomycin and antimycin A (OA), another mitochondrial depolarizing stress (Sup-plementary Fig. 2a), reducing the CCCP concentration to 5 μM reveals the similar phenotype to that observed with oligomycin treatment, which is that DELE1 KD cells have significant higher levels of mitophagy (Fig. 1e, f).

Oligomycin induces mitophagy in only ~5–10% NTC cells, regard-less of PRKN expression levels, suggesting that PRKN is not a limiting factor for oligomycin-induced mitophagy in WT cells (Fig. 1c–f). Strikingly, 24 h oligomycin treatment induces robust mitophagy in DELE1 KD cells, reaching levels that surpass those induced by 10 μM CCCP in cells with low PRKN expression, and comparable to CCCP-induced mitophagy in cells with medium or high PRKN expression (Fig. 1e, f), albeit with slower kinetics (Fig. 1g). The enhanced mito-phagy is significantly inhibited by bafilomycin A1, an inhibitor of autophagy that blocks the H + -ATPases and autophagosome-lysosome fusion[37] (Supplementary Fig. 1c–g).

We further confirmed that DELE1 KD promotes mitophagy under oligomycin by immunoblotting for the autophagy marker LC3 and a mitochondrial protein COX IV. Following oligomycin treatment, DELE1 KD results in an obvious increase in LC3-II levels, the lipidated form

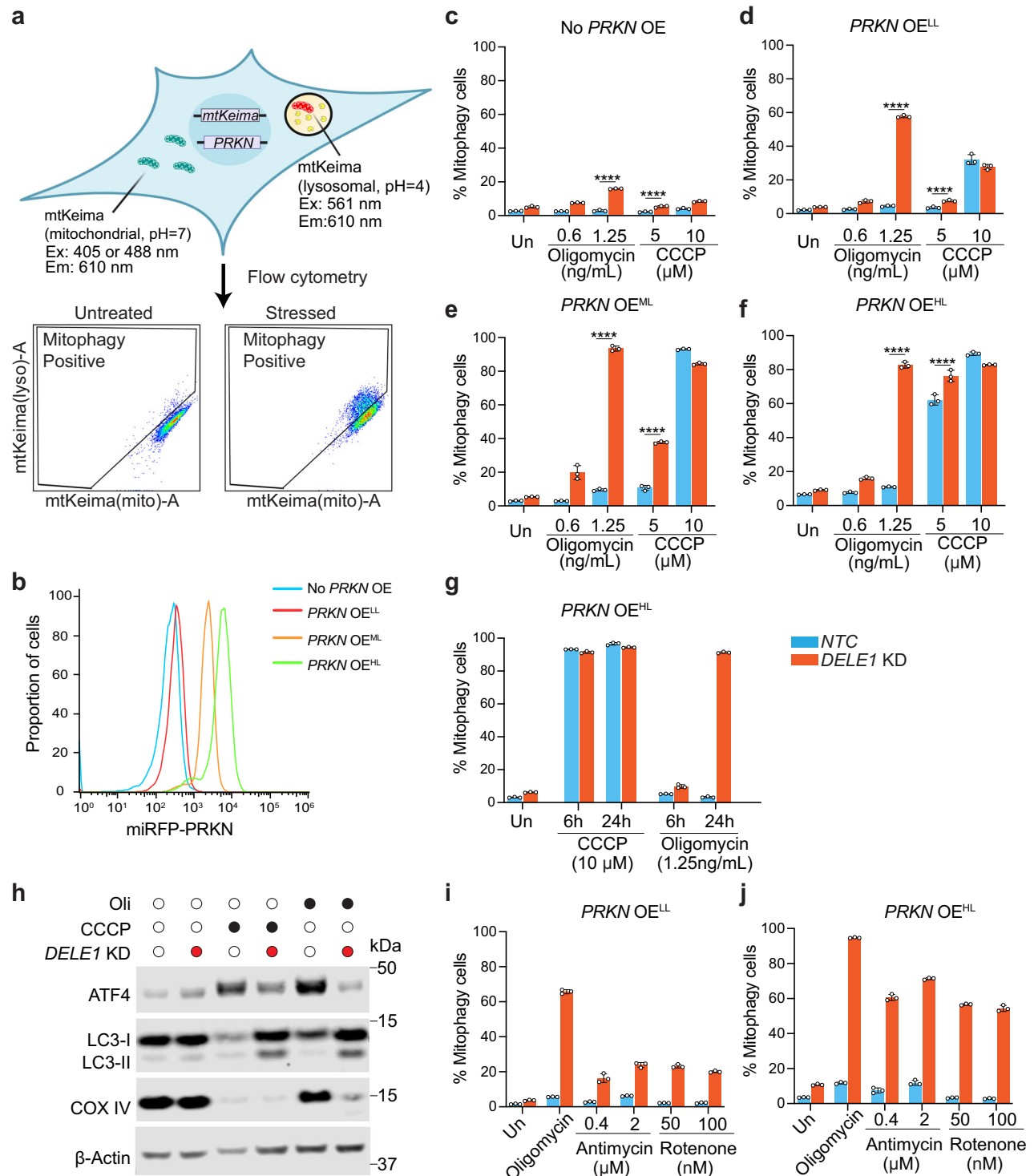

**Fig. 1 | DELE1 negatively regulates mitophagy under mild mitochondrial stress conditions. a** Schematic illustration for monitoring mitophagy using *mtKeima* reporter via flow cytometry. Cells with increased 561 nm/405 nm mtKeima ratios compared to non-stressed cells were considered as mitophagy-positive. Created in BioRender. Guo, X. (https://BioRender.com/jqwlsdj). **b** PRKN expression levels were measured using flow cytometry as indicated by miRFP intensity. **c** HEK293T *mtKeima* reporter cells without overexpression (OE) of *PRKN*, **d** with a lower level (*PRKN* OE^LL), **e** with a medium level (*PRKN* OE^ML), **f** with a high level (*PRKN* OE^HL), expressing a non-targeting control sgRNA (*NTC*) or sgRNA targeting *DELE1* (*DELE1* KD) were left untreated or treated with 5 or 10 µM CCCP, 0.6 or 1.25 ng/ml oligomycin, followed by measurement of mitophagy using flow cytometry. **c–f** show mean ± s.d., *n* = 3 independently treated culture wells, Two-way ANOVA test

followed by Turkey's multiple comparisons test (two-sided). **** adjusted *p* value < 0.0001. **g** *PRKN* OH^HL with or without *DELE1* KD were treated with 10 µM CCCP or 1.25 ng/ml oligomycin, followed by mitophagy measurement at 6 h and 24 h timepoints. (mean ± s.d., *n* = 3 independently treated culture wells) **h** Immunoblots of ATF4, LC3 and COX IV. *PRKN* OE^HL cells with or without *DELE1* KD were treated with 10 µM CCCP or 1.25 ng/mL of oligomycin for 24 h. β-actin was used as the loading control. **i** *PRKN* OE^LL and **j** *PRKN* OH^HL cells with or without *DELE1* KD were treated with antimycin A (0.4 and 2 µM) or rotenone (50 and 100 nM) for 24 hr, followed by measurement of mitophagy using flow cytometry. (mean ± s.d., *n* = 3 independently treated culture wells). Source data are provided as a Source Data file.

associated with autophagosome accumulation[38], indicative of enhanced autophagic activity. Consistently, COX IV levels are significantly reduced in *DELE1* KD cells (Fig. 1h), further supporting a negative regulatory role of DELE1 in mitophagy, particularly under mild mitochondrial stress conditions.

In addition to oligomycin, we observed increased mitophagy in *DELE1* KD cells under other mild stress conditions such as ETC Complex I inhibitor rotenone and Complex III inhibitor antimycin A in a PRKN-dosage dependent manner supported by both *mtKeima* reporter assays (Figs. 1i, j) and immunoblotting results (Supplementary Fig. 2b). Notably, oligomycin triggers the strongest mitophagy in *DELE1* KD cells among all OXPHOS inhibitors, prompting us to use oligomycin as the primary stressor in our subsequent mechanistic studies.

Iron deficiency can activate the DELE1-ISR[11] and induce mitophagy[39]. A recent study by Chakrabarty et al. reported that the DELE1 pathway promotes deferiprone (DFP, an iron chelator) and CCCP-induced mitophagy in HeLa and K562 cells[40]. In contrast, our findings in HEK293T show that DFP induces only mild mitophagy, which is slightly enhanced by *DELE1* KD in a PRKN-dosage-independent manner (Supplementary Fig. 2c and 2d). In HeLa cells, oligomycin also induced significantly higher levels of mitophagy in *DELE1* KD cells, similar to the effect observed in HEK293T cells (Supplementary Fig. 3a). In contrast, DELE1 did not promote DFP-induced mitophagy (Supplementary Fig. 3b). We also use K562, as in their study, and found that *DELE1* KD significantly enhances mitophagy across several stress conditions (Supplementary Fig. 3c). While their study suggested that the DELE1 pathway promotes mitophagy by recruiting phosphorylated eIF2α to mitochondria upon mitochondrial stress to promote mitophagy, our analysis did not support this. Specifically, extensive cellular fractionation under mitochondrial stress and other ISR-inducing stress showed no mitochondrial-specific enrichment of phospho-eIF2α (Supplementary Fig. 4a, b). Consistently, immunofluorescence staining of phospho-eIF2α revealed no mitochondrial stress-specific localization to mitochondria (Supplementary Fig. 4c). The reasons for these discrepancies remain unclear.

## DELE1 suppresses PINK1 stabilization in response to mild mitochondrial stress

Given that mitophagy induction in *DELE1* KD cells is much stronger in *PRKN* OE cells (Fig. 1), it is likely that oligomycin-induced mitophagy in *DELE1* KD cells occurs through the PINK1-PRKN pathway. To test this, we established *PINK1* knockout (KO) cell lines (validation in Supplementary Fig. 16) and measured mitophagy. Similar to CCCP or OA treatment, oligomycin-induced mitophagy in *DELE1* KD cells is abolished in the absence of PINK1 (Fig. 2a, Supplementary Fig. 2a). This requirement for PINK1 was also observed following treatment with antimycin, rotenone, and DFP (Supplementary Fig. 2a and 2e).

PINK1 accumulation upon mitochondrial depolarization triggers mitophagy. To examine whether DELE1 affects this process, we immunoblotted PINK1 in WT and *DELE1* KD cells with *PRKN* OE[ML] after treatment with oligomycin, antimycin A, or CCCP (5 or 10 μM). Consistent with previous studies, 10 μM CCCP robustly stabilizes full-length PINK1, whereas 5 μM results in only a mild increase in WT cells; this effect is further enhanced by *DELE1* KD. While oligomycin or antimycin A treatment for 24 h does not stabilize PINK1 in WT cells, PINK1 is significantly accumulated in *DELE1* KD cells (Fig. 2b). We further performed subcellular fractionation and found that PINK1 accumulates in the mitochondrial fraction in *DELE1* KD cells under both oligomycin and antimycin A conditions similarly to CCCP, although to a less extent (Fig. 2c, d).

Notably, although the PINK1 levels in *DELE1* KD cells following oligomycin treatment are significantly lower than those observed with 10 μM CCCP treatment in both WT and *DELE1* KD cells, they still reach a threshold level sufficient to trigger a comparable level of mitophagy

(Fig. 2b and Fig. 1f) possibly via a feedforward mechanism, albeit with slower kinetics (Fig. 1g)[41,42]. Coinciding with slower mitophagy relative to CCCP, oligomycin induces a delayed accumulation of PINK1 in *DELE1* KD cells (Supplementary Fig. 5). These findings, together with the *mtKeima* results, suggest that DELE1 negatively regulates mitophagy by modulating PINK1 stability. Interestingly, although the levels of PINK1 and phosphorylated ubiquitin levels are similar following treatment with oligomycin, antimycin A and 5 μM CCCP, mitophagy levels, measured by both *mtKeima* and immunoblotting for mitochondrial proteins (MFN1, MFN2, TIMM23, COX IV, SDHB) are notably higher with oligomycin than antimycin A or 5 μM CCCP. This may be due to previously reported autophagy inhibition by antimycin A[43] and CCCP[44].

To understand the mechanisms of PINK1 stabilization, we measured the mitochondrial membrane potential, as its loss is a known mechanism that stabilizes PINK1 and induces mitophagy[24]. Previous studies suggested that oligomycin can hyperpolarize mitochondria during the initial stage and eventually slightly depolarize the mitochondria[45]. Here, we detected oligomycin treatment for 24 h slightly reduces the membrane potential in WT (Fig. 2e). We did notice a small reduction of TMRE signal in *DELE1* KD cells following oligomycin treatment compared to WT cells, although the signal remained significantly higher than that observed with 10 μM CCCP treatment (Fig. 2d). We speculate this slight reduction maybe due to both minor depolarization and a smaller number of mitochondria, a result of slightly higher mitophagy in *DELE1* KD cells (Fig. 1c, and Supplementary Fig. 1a). Noticeably, membrane potential is comparable between WT and *DELE1* KD cells after 5 μM CCCP or rotenone treatment, yet DELE1 KD cells exhibited greater mitophagy under both conditions (Fig. 1e, i, j). Thus, the slight depolarization in *DELE1* KD cells after oligomycin treatment is not primary driver for PINK1 accumulation.

## DELE1 maintains mitochondrial protein import efficiency

Previous studies have shown that impairment of mitochondrial import machinery can stabilize PINK1 without loss of mitochondrial membrane potential[31,46]. Therefore, we hypothesize that DELE1 pathway may play a role in maintaining mitochondrial protein import under mild stress conditions. To investigate this, we conducted orthogonal assays to monitor mitochondrial presequence protein import.

First, we performed immunoblotting for HSPD1, a mitochondrial matrix protein. HSPD1 is translated as a precursor form containing a mitochondrial targeting sequence (MTS) that is cleaved upon successful import, and accumulation of the precursor indicates impaired mitochondrial import[31]. As expected, we observed a slight accumulation of HSPD1 precursor, indicating import failure, following CCCP treatment, but not oligomycin treatment in WT cells (Fig. 3a). Notably, *DELE1* KD cells exhibit more accumulation of the HSPD1 precursor, after both CCCP and oligomycin treatments (Fig. 3a). These results suggest that DELE1 may positively regulate mitochondrial import under both stress conditions. Consistently, we observed a further accumulation of PINK1 even under CCCP treatment in *DELE1* KD cells compared to WT cells (Fig. 2b). However, this increase in PINK1 level does not further enhance mitophagy, as CCCP treatment in WT cells already results in sufficient PINK1 accumulation to trigger mitophagy.

To exclude the possibility that the HSPD1 precursor and PINK1 accumulation following oligomycin treatment result from elevated translation in *DELE1* KD cells due to ISR inactivation rather than impaired import, we generated a cell line with inducible expression of a mitochondrial matrix protein, ornithine transcarbamylase (OTC), tagged with V5 at its C-terminus (OTC-V5), which can be induced concurrently with mitochondrial stress. This system enables quantitative assessment of the import efficiency for newly synthesized proteins during stress by calculating the ratio between the precursor and total OTC-V5. Treatment with CCCP leads to an exclusive accumulation of the precursor OTC-V5 with or without *DELE1* KD, indicating that

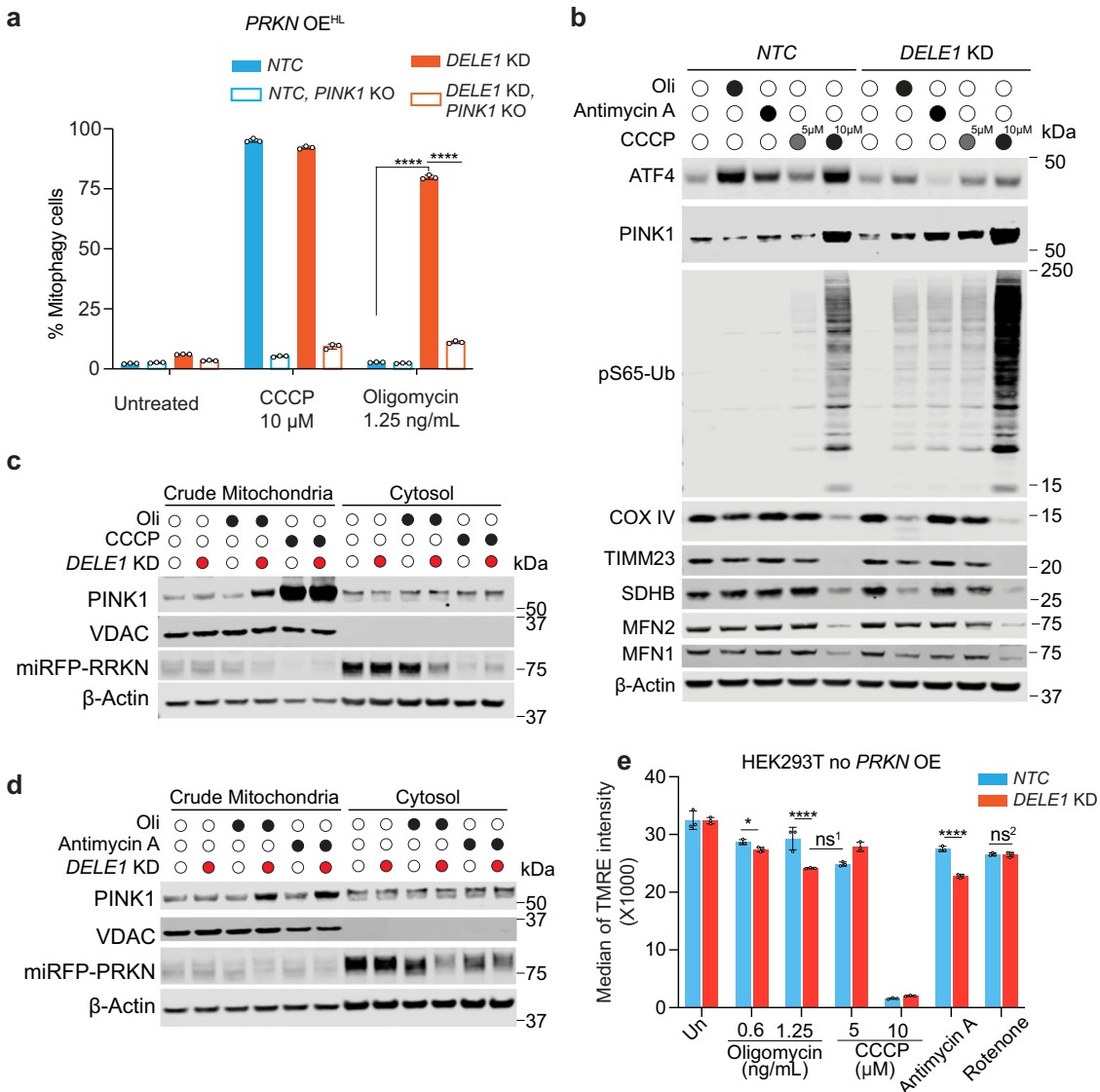

**Fig. 2 | Mild mitochondrial stress promotes PINK1 accumulation on mitochondria in *DELE1*-deficient cells. a** HEK293T *PRKN* OE^HL wild type (WT) or *PINK1* KO cells with or without *DELE1* KD were treated with 10 μM CCCP or 1.25 ng/ml oligomycin for 24 hr, followed by measurement of mitophagy using flow cytometry. (mean ± s.d., *n* = 3 independently treated culture wells) Two-way ANOVA test followed by Turkey's multiple comparisons test (two-sided). **** adjusted *p* value < 0.001. **b** Immunoblots of ATF4, PINK1, phosphorylated ubiquitin (pS65-Ub), LC3, COX IV, TIMM23, SDHB, MFN1, and MFN2. HEK293T *PRKN* OE^ML cells with or without *DELE1* KD were treated with either 5 or 10 μM CCCP, 1.25 ng/mL of oligomycin (Oli), or 100 nM antimycin A for 24 h. β-actin was used as the loading control. **c** Immunoblots of PINK1, PRKN, and VDAC. Mitochondrial and cytosolic fractions were isolated from *PRKN* OE^HL cells expressing an *NTC* or *DELE1* sgRNA. Cells were treated with 1.25 ng/mL oligomycin or 10 μM CCCP for 24 h before mitochondrial isolation. **d** Immunoblots of PINK1, PRKN, and VDAC. Mitochondrial and cytosol fractions were isolated from *PRKN* OE^HL cells expressing an *NTC* or *DELE1* sgRNA. Cells were treated with 1.25 ng/mL oligomycin (Oli) or 100 nM antimycin A for 24 h before mitochondrial isolation. **e** HEK293T cells with an *NTC* or *DELE1* sgRNA were treated with either 5 or 10 μM CCCP, 0.6 or 1.25 ng/ml oligomycin, 100 nM antimycin A, or 100 nM rotenone for 24 hr, followed by 100 nM TMRE staining and flow cytometry analysis. (mean ± s.d., *n* = 3 independently treated culture wells) Two-way ANOVA test followed by Turkey's multiple comparisons test (two-sided). ns, not significant, ns^1, adjusted *p* value = 0.996, ns^2, adjusted *p* value > 0.99, **** adjusted *p* value < 0.0001, * adjusted *p* value = 0.0475. Source data are provided as a Source Data file.

10 μM CCCP condition is a strong depolarizing stress that abolishes mitochondrial protein import. Following oligomycin treatment, *DELE1* KD cells exhibits significantly higher precursor-to-total protein ratios, a ~18-fold greater than those in WT cells (Fig. 3b), supporting a role for DELE1 in maintaining mitochondrial protein import under mild stress conditions. Because we still observe a strong accumulation of mature OTC-V5, it is possible that oligomycin-induced import defect requires a longer exposure, as suggested by the delayed accumulation of PINK1. To test this, we induced OTC-V5 expression after cells experienced varying durations of mitochondrial stress. Indeed, pretreatment with oligomycin significantly reduced mitochondrial import

(Supplementary Fig. 6a). In addition, the precursors of OTC-V5 remain associated with mitochondria and are accessible by proteinase K (Supplementary Fig. 6b), indicating failed import into mitochondrial matrix and raising a possibly that they occupy and further block the translocases. It is unlikely that precursor accumulation results from impaired degradation, since proteasome inhibition alone does not induce precursor buildup in WT (Supplementary Fig. 6c).

We further validated these findings using another inducible fluorescence reporter via microscopy-based approach. Specifically, we generated a reporter containing MTS from *COX8* fused to YFP (*MTS^COX8*-*YFP*). Expression of *MTS^COX8*-*YFP* was induced simultaneously with

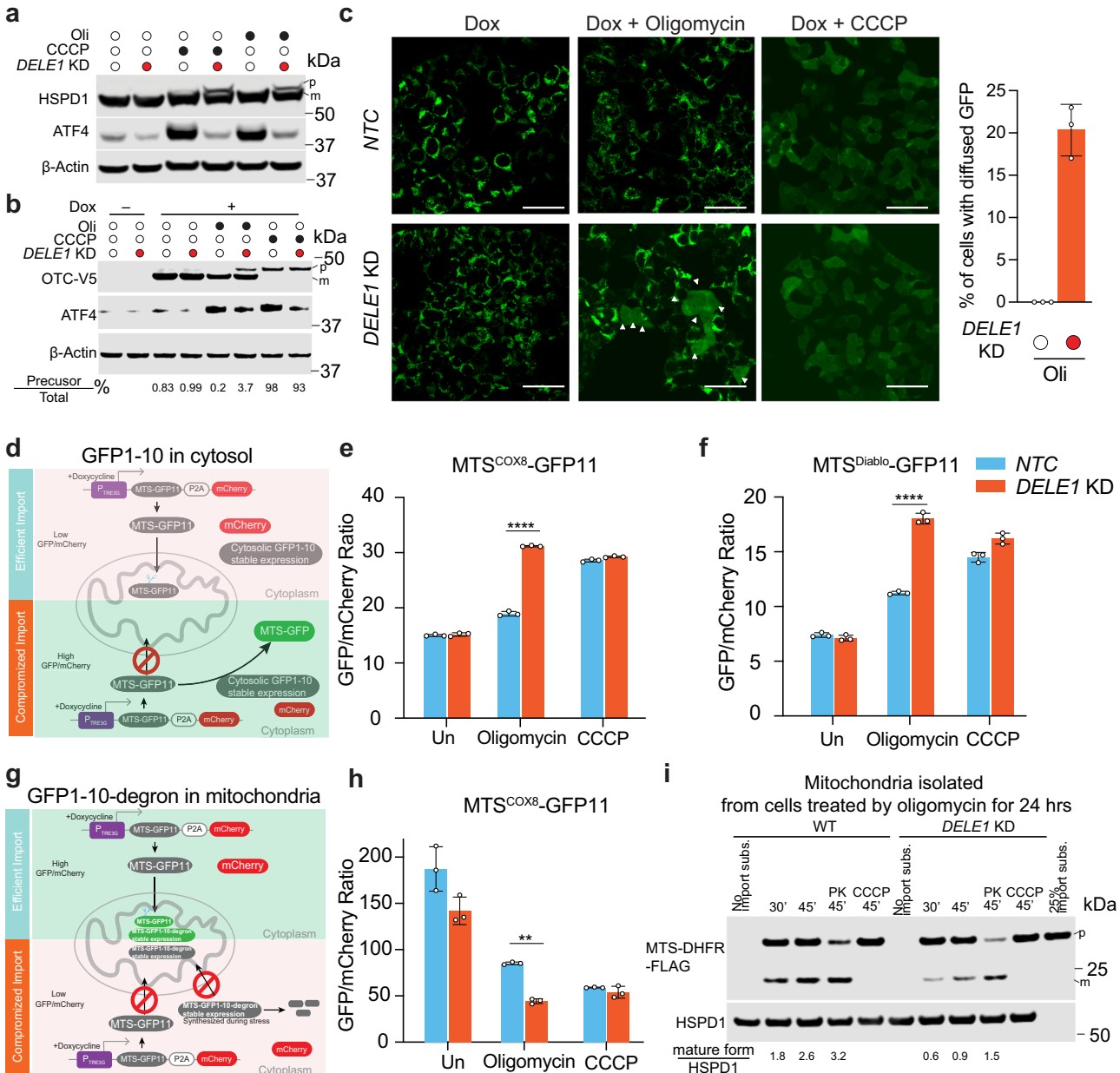

**Fig. 3 | DELE1 maintains mitochondrial presequence protein import upon stress. a** Immunoblots of HSPD1 and ATF4. HEK293T cells with an *NTC* or *DELE1* sgRNA (*DELE1 KD*) were treated with 10 µM CCCP or 1.25 ng/mL oligomycin (Oli) for 24 h. p: precursor form of HSPD1. m: mature form of HSPD1. β-actin was used as a loading control. **b** Immunoblots of OTC-V5 (via anti-V5 antibody) and ATF4. HEK293T inducible OTC-V5 cells with an *NTC* or *DELE1* sgRNA were treated 500 ng/mL doxycycline (Dox) concurrently with 10 µM CCCP or 1.25 ng/mL oligomycin (Oli) for 24 h. p: precursor form of OTC-V5. m: mature form of OTC-V5. β-actin was used as a loading control. Listed numbers in the bottom are the fractions of non-imported OTC-V5, calculated by dividing the level of precursor by the total (precursor + mature) protein level. **c** Left: Inducible *MTS^COX8^-YFP* cells with an *NTC* or *DELE1* sgRNA were treated with 500 ng/mL doxycycline (Dox) concurrently with 10 µM CCCP or 1.25 ng/mL oligomycin for 24 h. Arrowheads highlight cells with obvious diffused YFP signals. Scale bar 100 µm. Right: Quantification of percentage of cells with obvious diffused YFP signals following oligomycin treatment (mean ± s.d., *n* = 3, each with about 100 cells analyzed) **d** Schematic illustration for monitoring mitochondrial protein import using split-GFP (inducible MTS-GFP11+ constitutive GFP1-10) reporter via flow cytometry. **e** Measurement of the import of MTS^COX8^-GFP11 in GFP1-10 cells with an *NTC* or *DELE1* sgRNA following 1.25 ng/mL oligomycin or 10 µM CCCP treatments. Expression of MTS^COX8^-GFP11 was induced

simultaneously during mitochondrial toxins treatment by adding doxycycline. (mean ± s.d., *n* = 3 independently treated culture wells) Two-way ANOVA test followed by Turkey's multiple comparisons test (two-sided). **** adjusted *p* value < 0.0001. **f** Measurement of the import of MTS^Diablo^-GFP11 in HEK293T cells with an *NTC* or *DELE1* sgRNA. (mean ± s.d., *n* = 3 independently treated culture wells) Two-way ANOVA test followed by Turkey's multiple comparisons test (two-sided). **** adjusted *p* value < 0.001. **g** Schematic illustration for monitoring mitochondrial protein import using split-GFP (inducible MTS-GFP11+ constitutive MTS-GFP1-10-degron) reporter via flow cytometry. **h** Measurement of the import of MTS^COX8^-GFP11 in MTS-GFP1-10-degron cells with an *NTC* or *DELE1* sgRNA following 1.25 ng/mL oligomycin or 10 µM CCCP treatments. Expression of MTS^COX8^-GFP11 was induced simultaneously during mitochondrial toxins treatment by adding doxycycline. (mean ± s.d., *n* = 3 independently treated culture wells) Two-way ANOVA test followed by Turkey's multiple comparisons test (two-sided). ** adjusted *p* value = 0.0012. **i** Cell free mitochondrial protein import assay using mitochondrial isolated from WT and *DELE1* KD cells treated with oligomycin for 24 h. The mitochondria were incubated with MTS-DHFR-flag for the indicated time. To dissipate membrane potential, mitochondria were pre-incubated with CCCP for 5 min. Proteinase K (PK) was used to digest unimported substate post import reaction. Source data are provided as a Source Data file.

oligomycin or CCCP. As expected, CCCP abolishes mitochondrial localization of the reporter in both WT and *DELE1* KD cells. In contrast, oligomycin selectively disrupted mitochondrial localization of MTS[COX8]-YFP in *DELE1* KD cells. Notably, -20–25% of DELE1 KD cells display a diffused pattern of YFP (Fig. 3c), indicating impaired mitochondrial import of the reporter in the absence of DELE1.

Next, we used split-GFP strategies to assess mitochondrial protein import efficiency[47,48]. In this system, an inducible MTS-GFP11 is coexpressed with mCherry as expression control. Cells constitutively express either cytosolic GFP1-10 (Fig. 3d) or MTS-GFP1-10-degron (Fig. 3g). The degron destabilizes any unimported MTS-GFP1-10. Reconstitution of GFP depends on both mitochondrial protein import and GFP1-10 location. Impaired mitochondrial import leads to mislocalization of MTS-GFP11 to the cytosol, increasing GFP/mCherry ratio with cytosolic GFP1-10, whereas in the MTS-GFP1-10 system, import failure separates GFP11 from GFP1-10, lowering GFP/mCherry ratio. Using these complementary reporters, we confirmed reduced mitochondrial protein import efficiency in the absence of DELE1 (Fig. 3e–h). Import defects in *DELE1* KD cells were observed following treatment with antimycin A and rotenone but not DFP (Supplementary Fig. 6d). Notably, no reduction in GFP/mCherry signal was detected by flow cytometry when expressing MTS-GFP1-10 lacking a degron. This likely reflects failed import of newly synthesized MTS-GFP1-10 and MTS-GFP11 after oligomycin treatment in *DELE1* KD cells, resulting in reconstitution of GFP in the cytosol (Supplementary Fig. 7). Finally, we performed a cell-free mitochondrial protein import assay as recently described[49]. While mitochondria isolated from untreated WT and *DELE1* KD cells showed comparable protein import efficiency (Supplementary Fig. 8a), mitochondria isolated from oligomycin-treated *DELE1* KD cells exhibited significantly slower import of the substrate compared to WT (Fig. 3i). Interestingly, mitochondria from CCCP-treated cells retain some import activity, albeit at a reduced rate, which is further compromised by *DELE1* KD (Supplementary Fig. 8b).

In addition to pharmacological stress, we sought to manipulate mitochondrial import genetically by reducing TIMM8A rather than directly mutating translocases on the mitochondrial membrane. *Although* TIMM8A is an intermembrane space chaperon protein that facilitates the import and correct insertion of hydrophobic proteins[50], its loss may diminish translocase efficiency, thereby reducing matrix protein import. Using split-GFP reporter, we found that *TIMM8A* KD barely affects the import of MTS[COX8]-GFP11, possibly because TIMM8A does not play a role in the import of TIMM23 and TIMM22 in HEK293T cells[51]. Strikingly, mitochondrial import is compromised in *TIMM8A* and *DELE1* double KD compared to single KDs or WT cells, indicating that ISR activation helps maintain mitochondrial protein import in the absence of *TIMM8A* (Supplementary Fig. 9a). Indeed, we observed mild ISR activation in *TIMM8A* KD cells even without any additional stress (Supplementary Fig. 9b). We then measured mitophagy using the *mtKeima* reporter cell lines. *TIMM8A* KD does not induce mitophagy, but *TIMM8A*, *DELE1* double KD induce much higher mitophagy under untreated or following treatment with oligomycin at even a low concentration (0.3 ng/mL) (Supplementary Fig. 9c). Collectively, these findings highlight a critical role for DELE1 in maintaining mitochondrial protein import and suppressing mitophagy under stress conditions.

## ISR inhibits mitophagy by preserving mitochondrial protein import via reduced protein synthesis

Mitochondrial stress conditions, including both CCCP and oligomycin, trigger the OMA1-DELE1-HRI axis of the ISR, resulting in reduced protein synthesis and increased ATF4 translation[9,10]. Next, we set out to determine if DELE1 maintains mitochondrial import and inhibits mitophagy through ATF4 upregulation downstream of OMA1, HRI and eIF2α phosphorylation. Similar to *DELE1* KD, loss of OMA1, which cleaves DELE1 in response to oligomycin, significantly enhances

oligomycin-induced mitophagy (Fig. 4a). However, the effect is notably smaller than that observed with *DELE1* or *HRI* KD, potentially due to partial ISR activation via residual OMA1 or alternative ISR pathways (Supplementary Fig. 10a). *HRI* KD (Fig. 4a) cells and cells deficient in eIF2α phosphorylation (*eIF2α*[S49/52/A]) (Fig. 4c, Supplementary Fig. 16) both exhibit strong mitophagy upon oligomycin treatment. At the same time, we observed mitochondrial import impairment and PINK1 accumulation in mitochondrial fractions in *HRI* KD (Fig. 4b, and Supplementary Fig. 10d) and *eIF2α*[S49/52/A] cells (Fig. 4d, and Supplementary Fig. 10b–e) upon oligomycin treatment. In contrast, *ATF4* KO cells do not show increased mitophagy (Fig. 4e) nor impair mitochondrial protein import (Fig. 4f, Supplementary Fig.10b, c and 10g) following oligomycin treatment. Additionally, measurements of mitochondrial membrane potential in cells lacking different DELE1-ISR components revealed no significant loss of membrane potential following oligomycin treatment (Supplementary Fig. 10h, i). These results suggest that the OMA1-DELE1-HRI-eIF2α axis regulates mitochondrial protein import efficiency and mitophagy independently of ATF4 upregulation, likely by attenuating general protein synthesis.

Overexpression of mitochondrial proteins, particularly those harboring a bipartite targeting signal, can cause mitochondrial import defects by overwhelming the import machineries in yeast and mammals[52,53]. We hypothesize that, in the absence of the ISR, because protein synthesis rate is not attenuated, the amount of mitochondrial protein precursors synthesized becomes overwhelming for stressed mitochondria, causing mitochondrial import deficiency. This import deficiency, in turn, stabilizes PINK1 and enhances mitophagy. To test this hypothesis, we treated cells with the translation inhibitor cycloheximide (CHX) in ISR-deficient cells to reduce protein synthesis and examined its effects on mitophagy and mitochondrial protein import.

Both WT and *DELE1* KD cells were treated with varying concentrations of CHX concurrently with either CCCP or oligomycin. For these experiments, we selected sub-optimal concentrations of CHX compared to those typically used in pulse-chase experiments, given that PINK1 stabilization requires ongoing mRNA translation[23]. As expected, high doses of CHX inhibited mitophagy in both WT and *DELE1* KD cells treated with CCCP (Fig. 4g), likely due to strong suppression of protein synthesis that limits PINK1 availability. Notably, oligomycin-induced mitophagy in *DELE1* KD cells showed significantly higher sensitivity to CHX compared to CCCP-induced mitophagy. The IC$_{50}$ of CHX required for mitophagy inhibition in *DELE1* KD cells under oligomycin treatment was 0.04 µg/mL, -575-fold lower than the 23 µg/mL under CCCP treatment. This striking difference in CHX sensitivity suggests that the inhibition of oligomycin-induced mitophagy in *DELE1* KD cells is unlikely solely result from reduced PINK1 translation. Rather, it raises the possibility that mitochondrial protein import is restored under these conditions. Consistent with this, using both versions of our split-GFP reporter, we found that CHX and oligomycin cotreatment significantly improved import efficiency in *DELE1* KD cells (Fig. 4h, and Supplementary Fig. 11). A cell-free mitochondrial import assay further confirmed that co-treatment significantly alleviates the import deficiency observed in *DELE1* KD cells following oligomycin treatment (Fig. 4i). Furthermore, HSPD1 precursor was significantly reduced in *DELE1* KD cells after cotreatment, together with PINK1 destabilization and mitophagy inhibition (Fig. 4j, and Supplementary Fig. 12a, top panel).

Next, we investigated whether mTOR inhibition, which also attenuates protein translation, could suppress mitophagy in ISR-deficient cells. WT and *DELE1* KD cells were co-treated with Torin1, a potent mTOR inhibitor, alongside oligomycin or CCCP. Although Torin1 by itself promotes general autophagy[54] and hence slightly enhances mitophagy (Supplementary Fig. 12b), it significantly reduces oligomycin-induced mitophagy in *DELE1* KD cells, but has no effect on CCCP-induced mitophagy, as shown by the *mtKeima* reporter assays (Fig. 4k). These phenotypes were also observed with rapamycin,

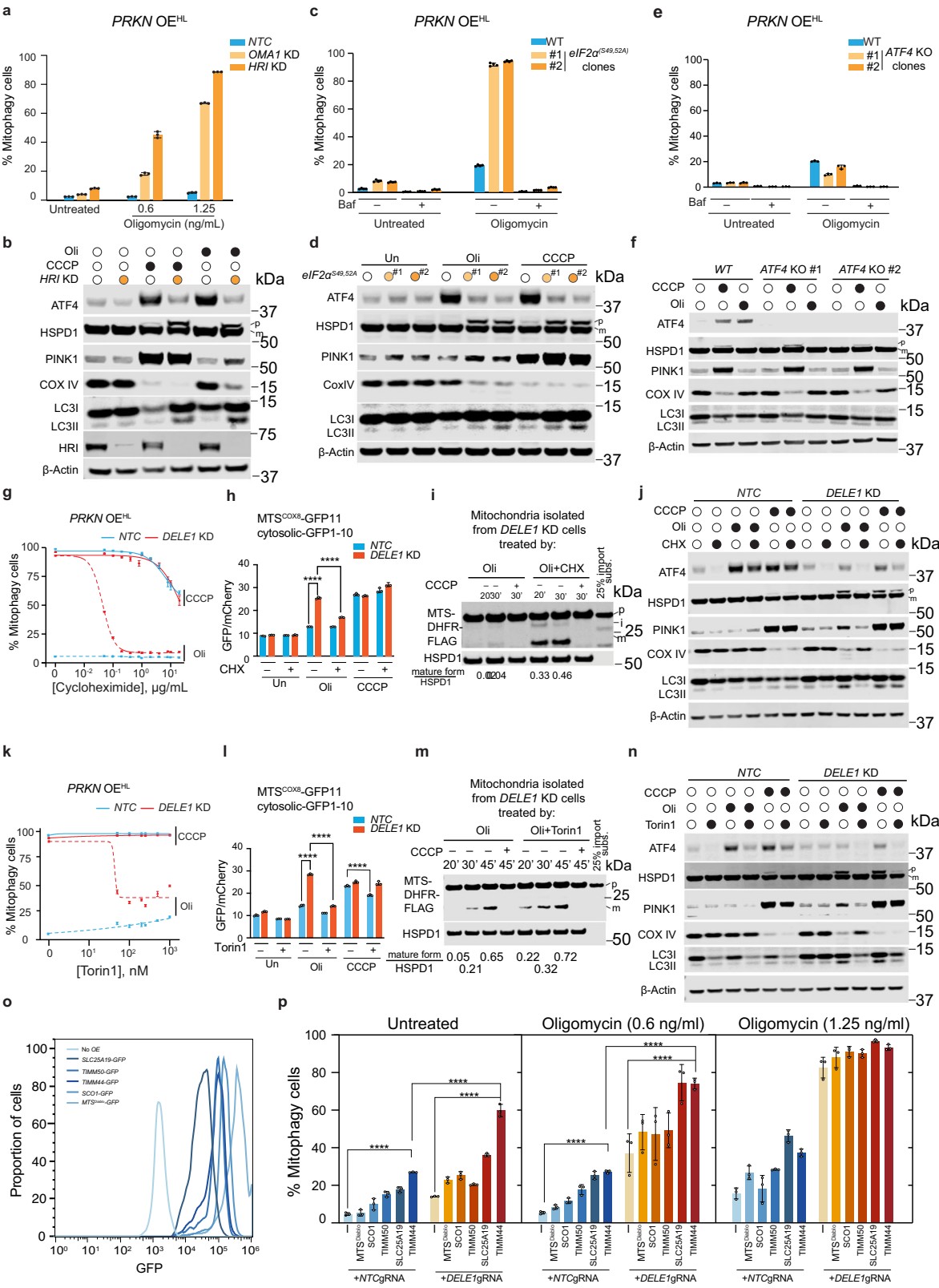

another mTOR inhibitor (Supplementary Fig. 12c-d). Similar to CHX, Torin1 improves mitochondrial protein import as shown by both split-GFP import reporters and cell-free mitochondrial import assays (Fig. 4l, m, and Supplementary Fig. 11). Consistent with the *mtKeima* assay, immunoblotting results showed elevated COX IV levels in *DELE1* KD cells co-treated with oligomycin and Torin1, along with PINK1 destabilization and reduction of the HSPD1 precursor (Fig. 4n,

Supplementary Fig. 12a, bottom panel). Interestingly, Torin1 co-treatment with CCCP also resulted in a notable reduction of PINK1 levels in WT (Fig. 4n). This reduction is unlikely to result from translation inhibition, as PINK1 was not identified as mTOR targets[55]. To evaluate Torin1's effects on PINK1 translation, epoxomicin, a proteasome inhibitor, was used to stabilize the cleaved PINK1, the levels of which serve as a proxy for PINK1 translation (Supplementary

**Fig. 4 | DELE1-ISR pathway negatively regulates mitophagy via attenuating protein synthesis. a** HEK293T *PRKN* OE^HL cells expressing a NTC sgRNA, or a sgRNA targeting *OMA1* (*OMA1* KD) or *HRI* (*HRI* KD) were left untreated or treated with 0.6 or 1.25 ng/mL oligomycin for 24 hr, followed by measurement of mitophagy via flow cytometry. (mean ± s.d., *n* = 3 independently treated culture wells) **b** Immunoblots of ATF4, HSPD1, PINK1, COX IV, LC3, and HRI. *NTC* and *HRI* KD cells with *PRKN* OE^HL were left untreated or treated with 10 μM CCCP or 1.25 ng/mL oligomycin for 24 h. β-Actin serves as loading control. **c** Wild type (WT) or two clonal *eIF2α^S49/52/A* cells with *PRKN* OE^HL were left untreated or treated with 1.25 ng/mL oligomycin with or without 100 nM bafilomycin A (Baf), followed by measurement of mitophagy via flow cytometry. (mean ± s.d., *n* = 4 independently treated culture wells) **d** Immunoblots of ATF4, HSPD1, PINK1, COX IV, LC3, phosphorylated and total eIF2α in WT and *eIF2α^S49/52/A* cells following 1.25 ng/mL oligomycin for 24 h. β-Actin serves as the loading control. **e** Flow cytometry measurement of mitophagy in WT or two *ATF4* KO clonal cell lines with *PRKN* OE^HL following treatment with 1.25 ng/mL oligomycin for 24 h in the presence or absence of bafilomycin A (Baf). (mean ± s.d., *n* = 3 independently treated culture wells) **f** Immunoblots of ATF4, HSPD1, PINK1, COX IV and LC3 and β-Actin in WT and two *ATF4* KO clonal cell lines following 10 μM CCCP or 1.25 ng/mL oligomycin treatment for 24 h. β-Actin serves as loading control. **g** *NTC* and *DELE1* KD cells with *PRKN* OE^HL were treated with 10 μM CCCP or 1.25 ng/mL oligomycin in the presence of cycloheximide at 12 different concentrations (0, 0.05, 0.1, 0.2, 0.4, 0.8, 1, 2, 4, 8,10 and 20 μg/mL) for 24 h followed by flow cytometry to measure mitophagy. Cycloheximide concentrations were converted to their base-10 logarithmic values. A nonlinear regression analysis using a log(inhibitor) vs. response model with a variable slope (four parameters) was performed to generate the plot. (mean ± s.d., *n* = 3 independently treated culture wells) **h** Measurement of the import of MTS^COX8-GFP11 in HEK293T cells with an *NTC* or *DELE1* sgRNA (*DELE1* KD) following treatment with 1.25 ng/mL oligomycin or 10 μM CCCP in the presence or absence of 100 ng/mL cycloheximide (CHX). MTS^COX8-GFP11 was induced simultaneously during mitochondrial toxins by adding 500 ng/mL doxycycline. (mean ± s.d., *n* = 3 independently treated culture wells) Two-way ANOVA test followed by Turkey's multiple comparisons test (two-sided). **** adjusted *p* value < 0.0001. **i** Cell free mitochondrial protein import assay using mitochondrial isolated from *DELE1* KD cells treated with oligomycin alone or cotreated with oligomycin and cycloheximide for 24 h. **j** Immunoblots of ATF4, HSPD1, PINK1, COX IV and LC3 and β-Actin in WT and *DELE1* KD cells with *PRKN* OE^HL following 10 μM CCCP or 1.25 ng/mL oligomycin treatment for 24 h with or without 100 ng/mL cycloheximide (CHX). p precursor; m mature. β-Actin serves as the loading control. **k** *NTC* or *DELE1* KD cells with *PRKN* OE^HL are treated with 10 μM CCCP or 1.25 ng/mL oligomycin in the presence of Torin1 at 7 different concentrations (0, 50, 100, 200, 250, 500 and 1000 nM) for 24 h followed by flow cytometry to measure mitophagy. Torin1 concentrations were converted to their base-10 logarithmic values. A nonlinear regression analysis using a log(inhibitor) vs. response model with a variable slope (four parameters) was performed to generate the plot. (mean ± s.d., *n* = 3 independently treated culture wells) **l.** Measurement of the import of MTS^COX8-GFP11 in HEK293T cells with an *NTC* or *DELE1* sgRNA (*DELE1* KD) following treatment with 1.25 ng/mL oligomycin or 10 μM CCCP in the presence or absence of 250 nM Torin1. MTS^COX8-GFP11 was induced simultaneously with mitochondrial toxins by adding 500 ng/mL doxycycline. (mean ± s.d., *n* = 3 independently treated culture wells) Two-way ANOVA test followed by Turkey's multiple comparisons test (two-sided). **** adjusted *p* value < 0.0001. **m.** Cell free mitochondrial protein import assay using mitochondrial isolated from *DELE1* KD cells treated with oligomycin alone or cotreated with oligomycin and Torin1 for 24 h. **n** Immunoblots of ATF4, HSPD1, PINK1, COX IV and LC3 and β-Actin in *NTC* and *DELE1* KD cells with *PRKN* OE^HL following 10 μM CCCP or 1.25 ng/mL oligomycin treatment for 24 h with or without 250 nM Torin1. p precursor; m mature. β-Actin serves as the loading control. **o** Measurement of the expression levels of different mitochondrial substrates fused with GFP at their c-terminus. **p** *mtKeima* reporter cells with or without overexpression of different mitochondrial substrates (MTS^Diablo-GFP, SCO1-GFP, TIMM50-GFP, SLC25A19-GFP, TIMM44-GFP), expressing an *NTC* or *DELE1* sgRNA were left untreated or treated with 0.6 ng/ml or 1.25 ng/ml oligomycin, followed by measurement of mitophagy using flow cytometry. (mean ± s.d., *n* = 3 independently treated culture wells) Two-way ANOVA test followed by Turkey's multiple comparisons test (two-sided). **** adjusted *p* value < 0.0001. Source data are provided as a Source Data file.

Fig. 12e–g). Surprisingly, Torin1 even slightly increases the cleaved PINK1, indicating PINK1 translation is not suppressed by Torin1. We also observed a reduction in ATF4 following Torin1 treatment (Fig. 4n), in line with previous reports that mTOR activity is required for ATF4 translation independent of eIF2α phosphorylation[56]. However, since ATF4 does not regulate mitophagy or mitochondrial import (Fig. 4e, f), Torin1 and ISR may have an additive effect on protein translation, improving mitochondrial import under stress conditions. Although PINK1 levels are reduced in WT or *DELE1* KD cells following co-treatment with CCCP and Torin1, they remain sufficient to trigger mitophagy (Fig. 4k, n).

These findings indicate that mild stress, such as oligomycin, increases the sensitivity of the mitochondrial import machinery to protein synthesis rates. When the ISR is inhibited, the mitochondrial import machinery becomes overwhelmed by mitochondrial substrates. Attenuating protein synthesis, either through general translation inhibition or mTOR inhibition, can alleviate mitochondrial import stress and suppress mitophagy. In contrast, under strong depolarizing stress, the mitochondrial import machinery is more severely compromised and less responsive to changes in protein synthesis rates.

To further support this model, we ectopically overexpressed various mitochondrial substrates, hypothesizing that overloading the mitochondrial translocases with substrates exhibiting slower import kinetics would enhance mitophagy, similar to the phenotype observed in *DELE1* KD cells under mild stress. We selected several candidates for overexpression, including the inner membrane space-localized MTS^Diablo-GFP, a peripheral inner membrane-associated protein, TIMM44, the inner membrane proteins TIMM50, SCO1 and SLC25A19. Interestingly, these mitochondrial substrates showed differential effects on mitophagy induction. Despite the highest expression level among all, as indicated by GFP intensities (Fig. 4o), the MTS^Diablo-GFP have minimal effect on mitophagy, suggesting an efficient import of

Diablo. In contrast, TIMM44 is the most potent candidate, capable of promoting mitophagy even under basal conditions, followed by SLC25A19 and TIMM50 (Fig. 4p). Their ability to induce mitophagy may reflect relatively inefficient import into mitochondria as indicated by the accumulation of precursor form of TIMM44 even without stress (Supplementary Fig. 13), in contrast, overexpressed MTS^Diablo-GFP can be imported efficiently. Notably, translation of TIMM44, TIMM50 and SLC25A19 is regulated by mTOR[55], mirroring the inhibitory effect of Torin1 on mitophagy observed in *DELE1* KD cells under stress. Finally, KD of *DELE1* further enhances mitophagy in these cell lines under both untreated and oligomycin conditions, suggesting DELE1 plays a positive role in maintaining mitochondrial protein import and suppressing mitophagy under import stress caused by mitochondrial protein overexpression.

## Modulating ISR using small molecules can regulate mitophagy

We next sought to investigate whether modulating the ISR through small-molecules inhibition or activation could regulate mitophagy. ISRIB, an ISR inhibitor that antagonizes phosphorylated eIF2α, promotes general protein translation while inhibiting ATF4 translation[32,33]. To investigate the effect of ISRIB on mitophagy induced by oligomycin, WT cells were treated with ISRIB at concentrations ranging from 10 nM to 2000 nM. Similar to *DELE1* KD, ISRIB significantly promotes mitophagy upon oligomycin treatment (Fig. 5a). However, the extent of mitophagy induction by ISRIB was less pronounced compared to *DELE1* KD. This difference may be due to ISRIB not fully blocking the ISR, as suggested by the ATF4 induction (Fig. 5b). As a result, only a small accumulation of HSPD1 precursor and PINK1 were observed (Fig. 5b). Interestingly, ISRIB exerts a stronger effect on CCCP treatment than oligomycin treatment, suggesting that unknown mechanisms associated with oligomycin may counteract the action of ISRIB.

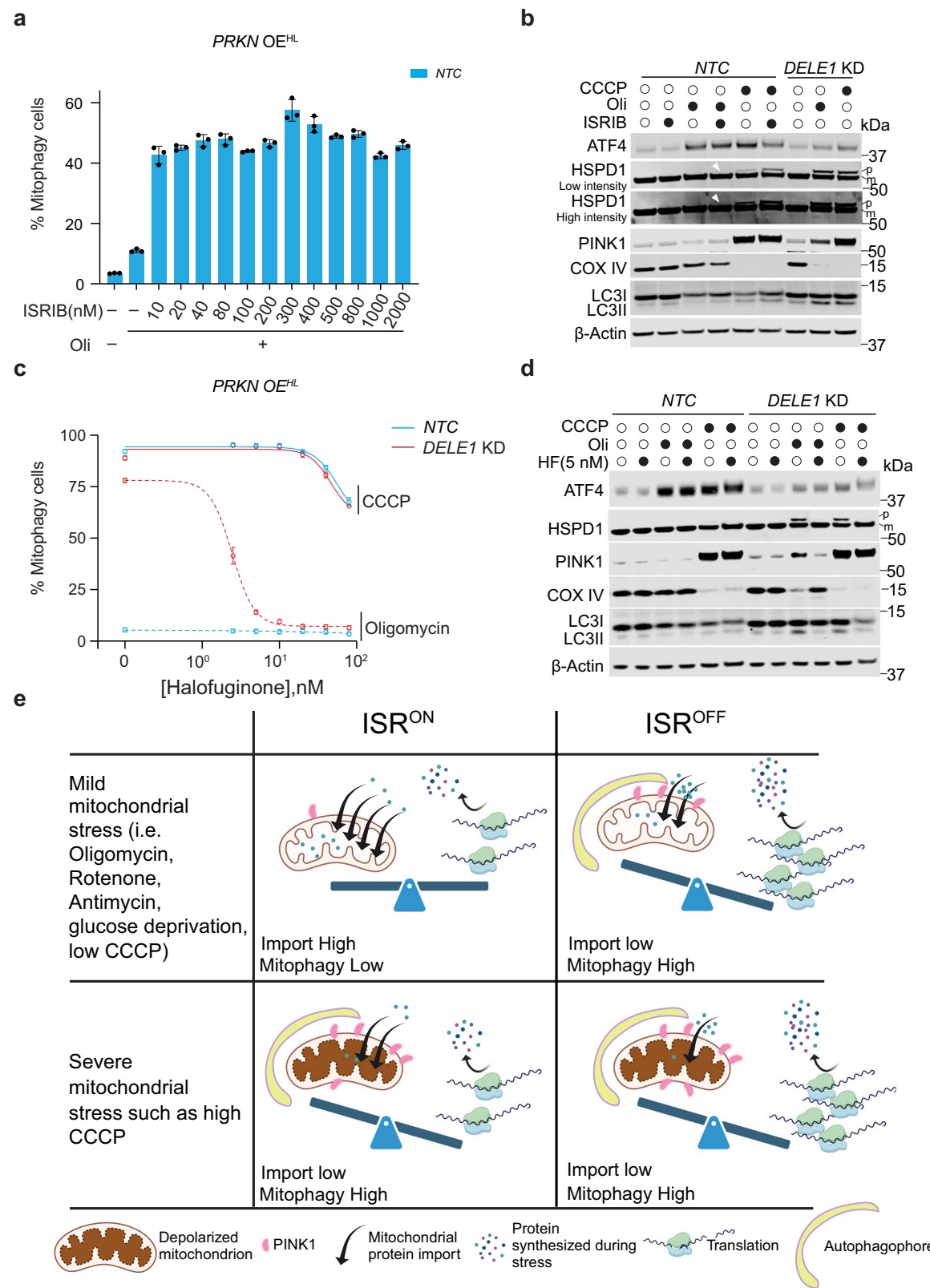

Next, we explored whether activating the ISR through alternative pathways could bypass the need for DELE1 in regulating mitochondrial import and mitophagy. Pharmacological activation of compensatory ISR kinases has been shown to mitigate cellular and mitochondrial dysfunction resulting from impaired PERK signaling[57]. Halofuginone (HF) induces the accumulation of uncharged proline tRNA, thereby activating the GCN2 branch of the ISR[58]. Strikingly, HF significantly suppresses mitophagy in *DELE1*-deficient cells treated with oligomycin, and this effect occurred at much lower concentrations than required for CCCP treatment (Fig. 5c). Co-treatment with 5 nM HF and oligomycin significantly reduces the accumulation of HSPD1 precursor and PINK1, indicating an enhanced mitochondrial import facilitated by HF, which is also supported by both versions of split-GFP import reporter (Supplementary Fig. 11 and 14a). This improved import leads to PINK1

**Fig. 5 | Modulation of mitophagy by small molecules that inhibit or activate the ISR. a** Measurement of mitophagy in WT HEK293T with *PRKN* OE^HL treated with 1.25 ng/mL oligomycin and cotreated with ISRIB at 13 different concentrations for 24 h. (mean ± s.d., *n* = 3 independently treated culture wells) **b** Immunoblots of ATF4, HSPD1, PINK1, COX IV and LC3 and β-Actin in *NTC* and *DELE1* KD cells with *PRKN* OE^HL following 10 μM CCCP or 1.25 ng/mL oligomycin treatment for 24 h with or without 300 nM ISRIB. p: precursor; m: mature. White arrow heads indicate a faint band of precursor form of HSPD1 in NTC cells following co-treatment with oligomycin and ISRIB. β-Actin serves as the loading control. **c** *NTC* and *DELE1* KD cells with *PRKN* OE^HL were treated with 10 μM CCCP or 1.25 ng/mL oligomycin in the presence of halofuginone at 7 different concentrations (0, 2.5, 5, 10, 20, 40 and 80 nM) for 24 h followed by flow cytometry to measure mitophagy. Halofuginone concentrations were converted to their base-10 logarithmic values. A nonlinear regression analysis using a log(inhibitor) vs. response model with a variable slope (four parameters) was performed to generate the plot. (mean ± s.d., *n* = 3 independently treated culture wells) **d** Immunoblots of ATF4, HSPD1, PINK1, COX IV and LC3 and β-Actin in *NTC* and *DELE1* KD cells with *PRKN* OE^HL following 10 μM CCCP or 1.25 ng/mL oligomycin treatment for 24 h with or without 5 nM halofuginone (HF). p precursor; m mature. β-Actin serves as the loading control. **e** Model for how the ISR pathway regulates protein translation, mitochondrial protein import, PINK1 stability and mitophagy under varying mitochondrial stress conations. Created in BioRender. Guo, X. (https://BioRender.com/v19u792).

destabilization and reduced mitophagy, as evidenced by the elevated levels of COX IV (Fig. 5d). Similarly, thapsigargin (THPG), a non-competitive inhibitor of the endoplasmic reticulum (ER) Ca²⁺ ATPase (SERCA)[59], which triggers ER stress and activates the PERK-mediated ISR pathway[60], also inhibits mitophagy in DELE1-deficient cells following oligomycin treatment (Supplementary Fig. 14b). However, THPG treatment at higher concentrations (>20 nM) exhibits high cytotoxicity, which may contribute to low mitophagy under both oligomycin and CCCP conditions at these concentrations. These results indicate that protein attenuation through alternative ISR pathways can preserve mitochondrial import efficiency under oligomycin treatment, which leads to PINK1 destabilization and mitophagy suppression (Fig. 5e).

## Optimal mitophagy is essential for cellular fitness

While impaired mitophagy has been implicated in numerous age-associated diseases due to the accumulation of dysfunctional mitochondria, excessive mitophagy can also be detrimental[61]. Over-activation of mitophagy may lead to metabolic failure and impaired function across various cell types, including neurons, cardiomyocytes, and stem cells. Thus, precise regulation of mitophagy is essential for maintaining cellular homeostasis under different physiological conditions[62].

Our previous studies indicate that the DELE1-mediated ISR is maladaptive in the context of oligomycin treatment[10]. To evaluate the role of mitophagy under oligomycin treatment, we compared the growth of both WT and *DELE1* KD cells with different levels of PRKN following treatment with 1.25 ng/mL oligomycin via crystal violet staining. In line with earlier findings, *DELE1* KD cells consistently grow better than WT cells, regardless of PRKN levels (Fig. 6a, Supplementary Fig. 15a). Strikingly, within the *DELE1* KD group, we observed a negative correlation between cell growth and PRKN levels (Fig. 6a). Since PRKN levels are positively associated with mitophagy, this suggests that mitophagy activation during oligomycin treatment may impair cell growth.

To further examine how dysregulated mitophagy may affect cellular fitness, we employed an alternative stress paradigm that forces cells to rely on mitochondria as their primary energy source by removing glucose from the culture medium. Similar to oligomycin treatment, albeit to a lesser extent, glucose deprivation activates the ISR, as evidenced by upregulation of ATF4 via DELE1 (Fig. 6b). Although PINK1 accumulation is minimal under these conditions, we observed a significant increase in mitophagy in *DELE1* KD cells cultured without glucose. This is likely due to elevated general autophagy in the absence of glucose, as indicated by increased LC3-II levels, in line with previous studies[63]. Mitophagy was confirmed by both the reduction of COX IV protein and *mtKeima* reporter assays. Similar to other mitochondrial stress conditions, glucose deprivation promotes mitophagy in *DELE1* KD cells in a manner dependent on both PRKN levels (Fig. 6c) and PINK1 (Fig. 6d).

Because glucose deprivation forces cells to reply on mitochondria for energy production, excessive mitophagy under these conditions may compromise cell fitness. To test this, we monitored cell growth in glucose-deprived medium over four days via crystal violet staining. These cells express different levels of PRKN, leading to different levels of mitophagy. Strikingly, opposite to observed with oligomycin treatment, *DELE1* KD cells grow worse than WT cells, regardless of PRKN levels (Fig. 6e, and Supplementary Fig. 15b). This is likely because that ATF4 activation, mediated by DELE1, may be necessary for cellular adaptation to the metabolic stress under glucose deprivation. To isolate the effect of mitophagy from other effects of the ISR such as ATF4 signaling, we next compared growth within *DELE1* KD group expressing different levels of PRKN. Among these, cells with low-level *PRKN* overexpression (*PRKN* OE^LL) show the best growth, outperforming cells with no *PRKN* overexpression as well as those with moderate or high *PRKN* expression levels (Fig. 6e, Supplementary Fig. 15b). These findings suggest that an optimal level of mitophagy is beneficial under glucose-deprived conditions, while insufficient or excessive mitophagy may impair cell survival.

## Discussion

Our studies uncover a link between the integrated stress response (ISR) activation and mitophagy under a wild range of mitochondrial stress conditions. The DELE1-HRI axis of the ISR is robustly triggered in response to most of mitochondrial stress conditions observed in many cell types, as well as in several mitochondrial myopathy mouse models[64–67]. In contrast, while PINK1-PRKN-dependent mitophagy is strongly activated by mitochondrial depolarization induced by CCCP, it occurs at significantly lower levels under other stress conditions, such as those induced by oligomycin. Through our investigation of ISR activation in relation to mitophagy induction, we identified a previously unrecognized role for ISR in negatively regulating PINK1-PRKN-dependent mitophagy.

Recently, two additional independent studies have uncovered a connection between the DELE1-HRI pathway and mitophagy, albeit with conflicting conclusions[40,68]. Chakrabarty et al., using whole-genome screening in K562 cells, identified a positive role for the DELE1-HRI pathway in mitophagy induced by iron deficiency[40]. Their findings suggest that the HRI branch of the ISR activation repurposes eIF2α phosphorylation from regulating translational initiation to initiating mitophagy in response to mitochondrial dysfunction. Furthermore, they concluded that HRI is broadly essential for mitophagy triggered by mitochondrial depolarization as well as hypoxia, as demonstrated in HeLa cells. Conversely, Singh, Agarwal, Volpi et al., found that the DELE1-HRI pathway negatively regulates mitophagy[68]. Their study began with a genetic siRNA screen targeting all known human Ser/Thr kinases in HeLa cells, revealing that *HRI* knockdown increases the stabilization and activation of PINK1, as evidenced by enhanced ubiquitin Ser65 phosphorylation following mitochondrial depolarization. Additionally, they observed that inhibiting the ISR pathway, either by genetic knockdown of *DELE1* or *HRI*, or by using the small-molecule inhibitor ISRIB, amplifies PINK1-dependent mitophagy while having little effect on iron-deficiency induced mitophagy in multiple cell lines

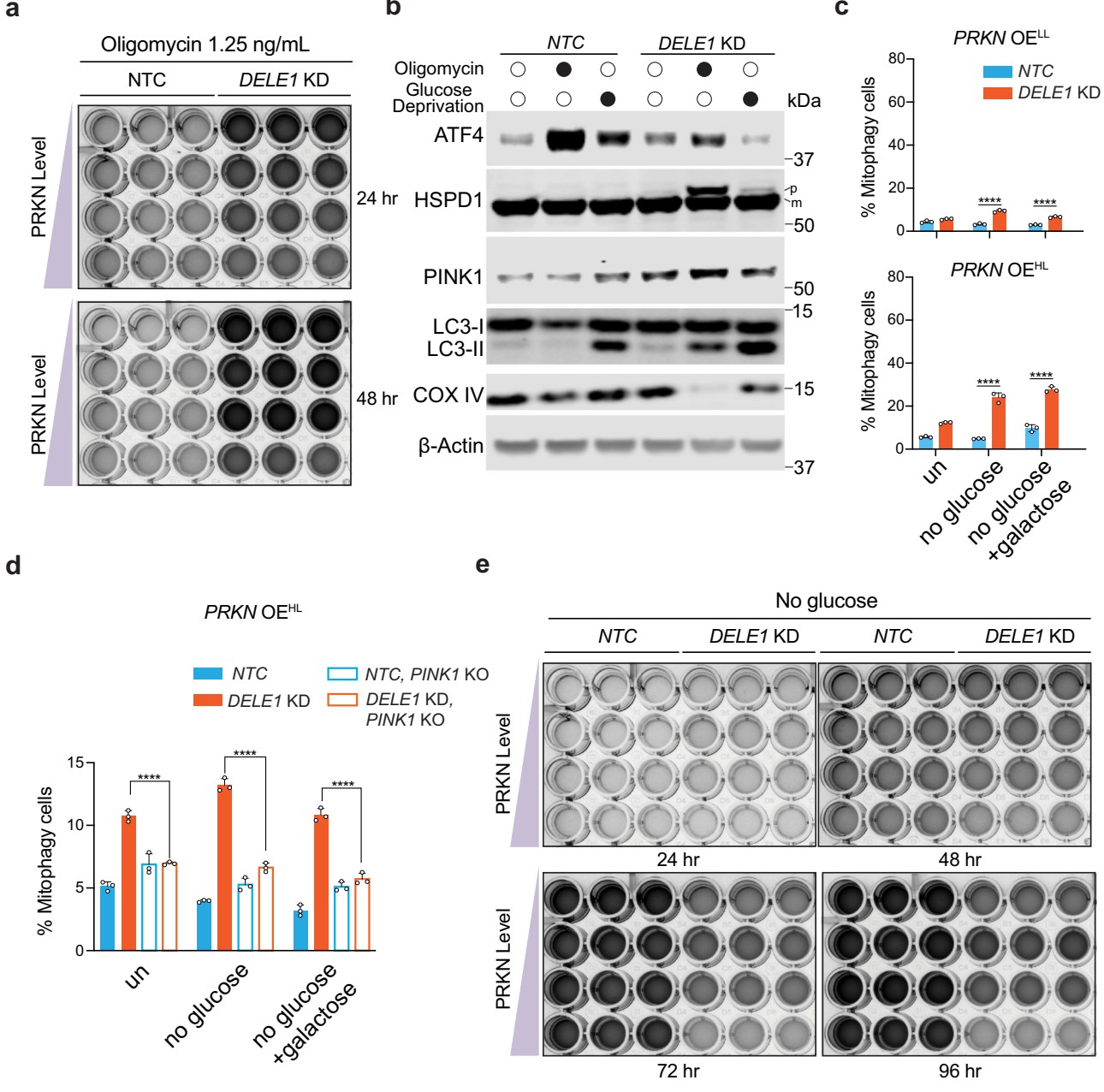

**Fig. 6 | Optimal mitophagy is essential for cellular fitness. a** Crystal violet staining of cells with different expression levels of PRKN with or without *DELE1* KD following treatment with 1.25 ng/mL oligomycin for 24 and 48 h. **b** Immunoblots of ATF4, HSPD1, PINK1, LC3 and COX IV in cells with *NTC* or *DELE1* KD cultured in medium with or without glucose for 24 h. p: precursor; m: mature. **c** *mtKeima* reporter cells with *PRKN* OE^LL or *PRKN* OE^HL, expressing an NTC or *DELE1* sgRNA (*DELE1* KD) were cultured with or without glucose (supplemented with galactose or not) for 24 h, followed by measurement of mitophagy using flow cytometry. (mean ± s.d., *n* = 3 independently treated culture wells) Two-way ANOVA test followed by Turkey's multiple comparisons test

(two-sided). **** adjusted *p* value < 0.0001. **d** *PRKN* OE^HL WT or *PINK1* KO cells with a *NTC* or *DELE1* sgRNA were cultured without glucose (with or without galactose) for 24 hr, followed by measurement of mitophagy using flow cytometry. (mean ± s.d., *n* = 3 independently treated culture wells) Two-way ANOVA test followed by Turkey's multiple comparisons test (two-sided). **** adjusted *p* value < 0.0001. **e** Crystal violet staining of cells with different expression levels of PRKN with or without *DELE1* KD cultured without glucose in the medium for 24, 48, 72 and 96 96 respectively. Source data are provided as a Source Data file.

they tested. The underlying reasons for these contrasting findings are not clear.

Similar to studies from Singh, Agarwal, Volpi et al., we provide strong evidence supporting a negative regulatory role for the ISR in mitophagy. We found that under a wide range of mitochondrial stress conditions that do not induce strong mitochondrial depolarization, PINK1-dependent mitophagy is significantly suppressed due to the activation of the DELE1-HRI-mediated ISR. However, under severe

depolarizing stress conditions, such as those induced by high concentration of CCCP and a combination of oligomycin and antimycin A (OA, as used in the aforementioned study[68], Supplementary Fig. 2), the absence of the ISR does not further enhance mitophagy. This is likely because mitophagy levels have already reached their maximum in the presence of the ISR, despite the observed increase in PINK1 stabilization in the absence of the ISR. The negative regulation on mitophagy by the ISR seems also occur in mouse. In several mouse models of

mitochondrial cardiomyopathy, outer mitochondrial membrane rupture and mitophagy were exclusively observed in the absence of *DELE1*[67]. Future studies are needed to determine whether mitophagy regulation in these mouse models contributes to the protective role of the DELE1 pathway in cardiomyopathy.

Our work reveals, for the first time, a mechanistic link connecting the ISR, protein synthesis, mitochondrial protein import and mitophagy. The negative regulation of mitophagy by the ISR is linked to its role in preserving mitochondrial presequence protein import efficiency. Mitochondrial protein import efficiency is intricately coupled to the regulation of mitochondrial homeostasis[69–71]. Although various stressors disrupt mitochondrial function through distinct mechanisms, they generally affect mitochondrial protein import efficiency to different extents. Among mitochondrial proteins, DELE1 appears particularly sensitive to import defects due to its unusually long mitochondrial targeting sequence[72], allowing it to sense a wide range of stress conditions and trigger the ISR[9,10,73]. In *C. elegans*, ATFS-1 contains a less efficient MTS, enabling it to sense mitochondrial perturbations as well as mitochondrial biogenesis during development[74]. On the other hand, PINK1 import is primarily impacted by more severe import defects, such as those caused by mitochondrial depolarization[22,23]. Recent research has shown that iron chelation halts DELE1 import, stabilizing full-length DELE1 on the outer mitochondrial membrane. Interestingly, this stress has minimal effect on mitochondrial presequence import measured by our split-GFP reporter (Supplementary Fig. 6d) nor it affects PINK1 import and stabilization[11]. Our findings here demonstrate that mitochondrial presequence import can be positively modulated by ISR activation under various pharmacological perturbations, as well as in response to mitochondrial DNA breaks, as shown in studies by Fu et al[75]. Future studies are required to elucidate whether the ISR affects additional mitochondrial import pathways during cellular stress.

The ISR regulates mitochondrial presequence protein import in an ATF4-independent manner, but it relies on the rate of protein synthesis. When the ISR is inactivated, increased protein translation leads to elevated levels of mitochondrial precursor proteins, which can overwhelm the mitochondria import machinery of stressed mitochondria and exacerbate protein import defects. Indeed, we observed compromised mitochondrial protein import in cells lacking the ISR pathway using multiple orthogonal approaches. A recent CRISPR screening study identified that the loss of genes involved in mitochondrial import machinery induces PINK1-dependent mitophagy, even without the loss of mitochondrial membrane potential[31]. Consistent with this, our study demonstrated that PINK1 becomes stabilized on the outer mitochondrial membrane to trigger mitophagy in mitochondrial-import-deficient cells due to an inactivation of the ISR following the treatment with various mitochondrial drugs as well genetically KD protein involved in protein import (i.e., TIMM8A) or OE of some mitochondrial proteins with slower import kinetics (i.e., TIMM44). Although we did not observe further enhancement of mitophagy following CCCP treatment, we did observe a further diminishment of mitochondrial import indicated by significant more accumulation of HSPD1 precursor, as well as stabilized PINK1 in the absence of DELE1 pathway compared to WT. We think this may help to reconcile with the finding that loss of HRI promotes PINK1 stabilization and mitophagy in HeLa cells following OA treatment.

Mitochondrial protein import is central to maintaining mitochondrial homeostasis[69]. Understanding how this process is regulated could pave the way for new strategies to modulate mitochondrial function and health. Our studies highlight a connection between the protein synthesis regulated by the ISR, mitochondrial protein import, and mitophagy (Fig. 5e). By mildly inhibiting protein synthesis with low doses of cycloheximide or mTOR inhibitors (Torin1 and rapamycin), in ISR-deficient cells, we were able to restore mitochondrial import and

suppress mitophagy. Pharmacological inhibition of the ISR with ISRIB in WT cells or activation of ISR in the absence of DELE1-mediated signaling either enhances or reduces mitophagy, respectively. Additionally, recent research indicates that ATAD1 supports mitochondrial import by extracting clogged precursor proteins resulting from import defects[53]. Future studies should investigate whether ATAD1 works synergistically with the ISR to protect mitochondrial import machinery and suppress mitophagy.

The ISR can play opposite roles on cellular fitness in a context-dependent manner[9,10,64,65,67]. Prolonged ISR activation is detrimental probably through upregulation of pro-apoptotic genes by ATF4, therefore, termination of ISR is critical and it is regulated by a large E3 ligase[47] and dephosphorylation of eIF2α[76]. In addition to ATF4 branch, our findings suggest that the ISR also regulates mitochondrial protein import and mitophagy. We propose a model in which stressed cells activate the ISR as a "first responder" to sustain mitochondrial import, prioritizing the transport of essential proteins necessary for mitochondrial repair rather than immediately triggering mitophagy. While defects in mitochondrial protein import play a critical role in sensing and responding to mitochondrial stress, they also pose a challenge, as homeostatic proteins must be efficiently imported into mitochondria to perform their repair functions after being upregulated in response to stress. In *C. elegans*, for example, the accumulation of misfolded proteins in mitochondria activates the mitochondrial unfolded protein response (UPR^mt) by reducing the import efficiency of ATFS1[77]. When ATFS1 fails to enter the mitochondria, it is relocated to the nucleus, where it acts as a transcription factor to upregulate mitochondrial chaperones and proteases, both of which need to be imported into mitochondria for proper repair. Interestingly, the UPR^mt can help preserve mitochondrial protein import by upregulating the mitochondrial import machinery[78]. In mammals, our studies suggest that ISR activation, through inhibition of protein synthesis, can support mitochondrial import under mild stress. Mitochondrial misfolding stress induced by GTPP (an Hsp90 inhibitor) triggers a mammalian UPR^mt by integrating mitochondrial import defects with oxidative stress, leading to the upregulation of mitochondrial homeostasis genes encoding mitochondrial chaperones and proteases[79,80]. Additionally, GTPP robustly activates the DELE1 axis of the ISR[79,80]. Although the ISR is not strictly required for UPR^mt induction, it is possible that ISR activation enhances mitochondrial import of upregulated mitochondrial chaperons, thereby helping to maintain mitochondrial homeostasis during stress. Indeed, our cell growth assays suggest that activation of mitophagy under both oligomycin and glucose deprivation play a negative role in cellular fitness, although ISR activation have opposite effects under these two conditions. This is likely due to other outcomes of ISR activation such as ATF4.

## Method

### Cell culture and cell lines

HEK293T cells (ATCC, CRL-3216) and HeLa cells (ATCC, CCL-2) were cultured in Dulbecco's modified Eagle's medium (DMEM) (Gibco, 11965-092). K562 (ATCC CCL-243) cell line was cultured in RPMI (Gibco, 22400-105). All cell culture media were supplemented with 10% fetal bovine serum (Seradigm, 1300-500), penicillin–streptomycin (Life Technologies, 15140122) and L-glutamine (Life Technologies, 25030081). Cells were cultured at 37 °C and 5% CO₂ in a humidified incubator.

The CRISPRi HEK293T cell line (cXG284) was generated previously[10]. The CRISPRi HeLa cell line was a gift from J. Weissman Lab. The *mtKeima* reporter cell line was generated through lentiviral infection of cXG284 with pMY004 (*CAG:mtKeima*) and FACS-based selection. The *miRFP-PRKN* overexpression cell lines were generated through lentiviral infection of the mtKeima cell line with pXG646 and FACS-based selection. The monoclonal cell lines with different expression levels of PRKN were generated and selected based on

the levels of miRFP. We also generated a polyclonal *PRKN* overexpression cell line via integration of *miRFP-PRKN* into the AAVS1 safe harbor locus[81]. Inducible *OTC-V5-P2A-mcherry* cell line was generated through lentiviral infection and FACS-based selection after induction by 500 ng/mL Doxycycline overnight. Inducible *MTS^COX8^-YFP* cell line was generated through lentiviral infection of pXG447 and FACS-based selection after induction by 500 ng/mL Doxycycline overnight.

*ATF4 (KO)* cell lines were generated by CRISPR-Cas9-mediated gene editing by using two sgRNAs targeting the first and second exons of *ATF4*. The protospacer sequences are: 5′-TCACCCTTG CTGTTGTTGGA-3′ and 5′-GCAGAGGATGCCTTCTC-3′ (Synthego, CA). *eIF2α^S49/S2/A^* cell lines were generated by CRISPR-Cas9-mediated gene editing through homologous recombination. The HEK293T cells were infected with a sgRNA with protospacer sequence 5′-TTCTTAGT-GAATTATCCAGA-3′, a donor ssDNA with a sequence of 5′-GTGAA TGTCAGATCCATTGCTGAAATGGGGGCTTATGTCAGCTTGCTGGAAT ACAACAACATTGAAGGCATGATTCTTCTTGCAGAATTAGCACGAAGA CGTATCCGTTCTATCAACAAACTCATCCGAATTGGCAGGAATGAGTG TGTGGTTGTCATTAGGGTGGACAAAGAAAAAGGTAAGTGAGAAAAAT-3′ and Cas9 protein via SNP delivery. *PINK1(KO)* cell lines were generated by CRISPR–Cas9-mediated gene editing by using two sgRNAs targeting the first exon of *PINK1*. The protospacer sequences are: 5′-TCTCCGCTTCTTCCGCCAGT-3′ and 5′-CCTCATCGAGGAAAAACAGG-3′ (cloned into pX459 from Addgene). Monoclonal *PINK1* KO were validated through PCR, immunoblot and phenotypic analysis. CRISPRi knockdown cell lines were generated by lentiviral transduction with plasmids containing individual sgRNAs and selected by puromycin at 2 μg/mL. Gene KD or KO was validated through WB or RT-qPCR (Supplementary Fig. 16). For split-GFP reporter cell lines, MTS (from *COX8* or *Diablo*) -GFP11 were integrated into the GFP1-10 cell line, which was generated by lentiviral transduction of pHR-SFFV-GFP1-10 (Addgene, #80409, a gift from Bo Huang's lab)[82] or MTS-GFP1-10 with or without degron generated in the lab. Mitochondrial proteins fused with GFP were stably integrated into the *mtKeima* reporter cell line and enriched via FACS sorting.

Cell lines were tested for mycoplasma contamination routinely with primers oXGmycop-F: 5′TGCACCATCTGTCACTCTGTTAACCTC3′ and oXGmycop-R: 5′GGGAGCAAACAGGATTAGATACCCT3′.

## Constructs

For constructing *mtKeima* reporter (pMY004), *mtKeima* was PCR amplified from pCHAC-mt-mKeima (Addgene, plasmid # 72342)[28] and cloned into a lentiviral vector pdbr3_CAG. For *PRKN* overexpression (pXG646), *PRKN* was PCR amplified from CFP-Parkin (Addgene, plasmid # 47560)[27] and cloned into a lentiviral vector with hPGK promoter and *miRFP*. For inducible *MTS-YFP* (pXG447), mitochondrial targeting sequence of *COX8* and *YFP* were cloned from *mito-SRAI* (#RDB18223)[83] into a lentiviral plasmid containing a TetON inducible promoter and trans-activator rtTA. For *PINK1* knock out, sgRNA protospacer sequences were cloned into pSpCas9(BB)-2A-Puro (PX459) V2.0 (Addgene, plasmid # 62988) following the protocol previously published[84]. For gene knockdown via CRISPRi, top and bottom oligonucleotides (IDT) were annealed and ligated to an optimized lentiviral sgRNA expression vector[85]. For cloning inducible *MTS^COX8^-GFP-11-P2A-mcherry (pXG655)*, *GFP11-P2A-mcherry* gene block was ordered from IDT and replaced YFP in plasmid pXG447. The MTS of *Diablo* (177 bp encoding for the first 59 amino acids) was synthesized from IDT and replaced MTS^COX8^ in pXG447 and pXG655 to generate *MTS^Diablo^-YFP* and *MTS^Diablo^-GFP-11-P2A-mcherry* respectively. The sequence of human TIMM44, TIMM50, SLC25A19, SCO1 were PCR amplified from HEK293T cell cDNA library. These DNA fragments was inserted into the NotI-BamHI site of pXG447 (inducible *MTS-YFP*). For inducible OTC-V5-P2A-mCherry, OTC is PCR amplified from the plasmid (addgene, #71877)[86] with V5 included in the primer and inserted into the NotI-BamHI site of pXG655. For MTS-GFP1-10 (pXG229), we introduced MTS^cox8^ into the plasmid pHR-SFFV-GFP1-10 (addgene, #80409)[82]. For MTS-GFP1-10-degron, we PCR amplified the CL1-PEST degron from *mito-SRAI* (#RDB18223)[83] and integrated it to the c-terminus of MTS-GFP1-10 via HiFi DNA assembly (NEB, E2621).

## Drug treatment

HEK293T cells and their derived cell lines were seeded at 25% confluency 24 h before drug treatment (i.e., 15,000 cells per well of a 96-well plate). An equal volume of DMEM with twice the final drug concentration was added. For mitochondrial stress, cells were incubated with the following mitochondrial toxins for 24 h unless otherwise stated: 1.25 ng/ml oligomycin (Sigma-Aldrich, 75351), 100 nM antimycin (Sigma-Aldrich, A8674), 100 nM, rotenone (Sigma-Aldrich, R8875), 10 μM carbonyl cyanide 3-chlorophenylhydrazone (CCCP) (Sigma-Aldrich, C2759), 1.25 ng/ml oligomycin combined with 100 nM antimycin (OA treatment) and 1 mM DFP (Sigma-Aldrich, 379409). Bafilomycin A1 (Cayman,11038) was used at a final concentration of 100 nM to suppress autophagy. A final concentration of 500 ng/mL of Doxycycline (Sigma-Aldrich, 631311) was used to induce gene expression. To attenuate protein synthesis, cycloheximide (Sigma-Aldrich, C7698) was used at a series of concentrations: 50 ng/mL, 100 ng/mL, 200 ng/mL, 400 ng/mL, 800 ng/mL, 1 μg/mL, 2 μg/mL, 4 μg/mL, 8 μg/mL,10 μg/mL and 20 μg/mL. mTOR inhibitor Torin1 (Cayman, 10997) was used at concentrations of 50, 100, 200, 250, 500 and 1000 nM. mTOR inhibitor Rapamycin (Cayman, 13346) was used at concentrations of 10, 25, 50, 100, 200, 250, and 500 nM. Epoxomicin (TargetMol Chemicals, T6830) was used at a concentration of 300 nM for 10 h to inhibit proteasome. To trigger the GCN2 branch of ISR, halofuginone (Sigma-Aldrich, 50-576-300001) was used at the following concentrations: 2.5, 5, 10, 20, 40, and 80 nM. To activate the PERK branch of ISR, thapsigargin (Sigma-Aldrich, T9033) was used at the following concentrations: 2.5, 5, 10, 20, 40, and 100 nM.

## Mitophagy measurement by flow cytometry

After 24 h of drug treatment, cells were digested with trypsin for 2-3 min at 37 °C and then trypsin was neutralized with four to five volumes of DMEM medium. Flow cytometry was performed on an Attune CytPix flow cytometer with an auto sampler (Invitrogen). Events were pre-selected for living and single cell populations. Excitation at 405 nm (pH 7) with 610/20 nm emission filters and excitation at 561 nm (pH 4) with 620/15 nm emission filters were selected to measure mtKeima fluorescence. The flow cytometry data was analyzed using the software FlowJo10.10.0 (FlowJo, LLC). Examples of gating strategies are shown in Supplementary Fig. 11. The bar graphs were generated using GraphPad Prism version 10 and Adobe Illustrator. Data were collected from three independently treated culture wells (biological replicates) analyzed on the same day.

## Mitochondrial Isolation

Cytosolic and mitochondrial fractions were separated using the Mitochondrial Isolation Kit for Mammalian Cells (Thermo Fisher Scientific, 89874). Mitochondrial pellet was lysed in RIPA buffer (Thermo Fisher Scientific, 89900) containing protease inhibitor cocktail (Millopore Sigma, 04693159001), followed by quantification of protein concentration using BCA assays (Thermo Fisher Scientific, 23227). 8 μg of total protein from each sample (both mitochondrial and cytosolic fractions) was loaded for immunoblotting experiments.

**Subcellular fractionation** (for phospho-eIF2α detection)
- Mechanical cell fractionation:

After CCCP or Thapsigargin treatment, cells were harvested with PBS containing 1 mM EDTA, then centrifuged at 750 g for 5 min, washed in PBS, and the cell pellet was resuspended in ice cold isotonic mitochondrial isolation buffer (MIB: 210 mM mannitol, 70 mM sucrose, 1 mM EDTA, and 10 mM Hepes (pH 7.5), supplemented with

protease inhibitor cocktail (Thermo Scientific). Cells were then lysed with a 27 ½–gauge syringe (BD Biosciences). Plasma membrane rupture of cells was confirmed by staining in a 0.2% trypan blue solution. Next, samples were transferred to Eppendorf centrifuge tubes and centrifuged at 1000 g for 5 min at 4 °C to eliminate nuclei and unbroken cells. The resulting supernatant was centrifuged at 10,000 g for 10 min at 4 °C to obtain the heavy membrane fraction. This supernatant was further centrifuged at 20,000 g for 30 at 4 °C to yield the light membrane pellet and the final soluble cytosolic fraction. Heavy membrane fraction was washed once in MIB then lysed in lysis buffer (50 mM Tris-HCl pH 7.4, 150 mM NaCl, 1 % Triton X-100, 2 mM EDTA, 2 mM sodium pyrophosphate, 25 mM β-glycerophosphate, 1 mM sodium orthovanadate) supplemented with protease inhibitor cocktail (Thermo Scientific). Cytosolic and heavy membrane fractions (10 mg per condition) were then analyzed by immunoblotting.

-Digitonin-based cell fractionation:

Cells were harvested with PBS containing 1 mM EDTA, then centrifuged at 750 g for 5 min, washed in PBS, and the cell pellet was digitonin-permeabilized for 5 min on ice after resuspension of the cell pellet in cytosolic extraction buffer (CEB: 250 mM sucrose, 70 mM KCl, 137 mM NaCl, 4.3 mM $Na_2HPO_4$, 1.4 mM $KH_2PO_4$ (pH 7.2), with 300 mg/ml digitonin, and the protease inhibitor cocktail. After 5 min incubation on ice, plasma membrane permeabilization of cells was confirmed by staining in a 0.2% trypan blue solution. Cells were then centrifuged at 1000 g for 5 min at 4 °C. The supernatants (cytosolic fractions) were saved and the pellets solubilized in the same volume of mitochondrial lysis buffer (MLB: 50 mM Tris, pH 7.4, 150 mM NaCl, 2 mM EDTA, 2 mM EGTA, 0.2% Triton X-100, 0.3% NP-40) supplemented with protease inhibitor cocktail, followed by centrifugation at 10,000 g for 10 min at 4 °C. After centrifugation, supernatants are kept as the heavy membrane fractions. Soluble and membrane fractions (10 mg per condition) were then analyzed by immunoblotting.

-Differential centrifugations:

Cells were mechanically disrupted with a 27 ½–gauge syringe in H60 buffer (20 mM HEPES pH 7.9, 1.5 mM $MgCl_2$, 60 mM KCl) supplemented with protease inhibitor cocktail. Samples were centrifuged at 1000 $g$ to remove the nuclei and unbroken cells. The supernatant (S1) was centrifuged at 5000 $g$ for 5 min, to precipitate heavy organelles (P5). The supernatant (S5) was further centrifuged at 10,000 $g$ for 10 min to generate S10 and P10. S10 was centrifuged at 25,000 × $g$ for 30 min to obtain the cytosolic fraction (S25) and P25. Each pellet was resuspended in lysis buffer and analyzed by immunoblotting.

## Immunoblotting

Cell samples were lysed with RIPA lysate (Thermo Fisher Scientific, 89900) containing protease inhibitor cocktail (Millopore Sigma, 04693159001) for 30 minutes on ice, and vortexed for 15 seconds every 10 minutes. The homogenate was centrifuged at 20,000 × g for 20 min at 4 °C, then the supernatant was collected. A BCA kit (Thermo Fisher Scientific, 23227) and a Spark multimode microplate reader (Tecan) was used to measure protein concentration. 20 μg proteins from each sample were separated by 4–12% gradient Bis-Tris Gels (Thermo Fisher Scientific) and transferred to 0.2 μm nitrocellulose membranes (Thermo Fisher Scientific, Catalog No. 77012). The membrane was blocked with 5% nonfat milk diluted in TBS (20 mM Tris-HCl, 150 mM NaCl, pH 7.5) for 1 h with gentle shaking at room temperature (RT), then incubated with the primary antibodies overnight (about 15 hr) at 4 °C. After three washes with TBS-T (TBS containing 0.1% Tween-20), the membrane was incubated with infrared fluorescent dye labeled secondary antibodies (LI-COR) for 2 h at RT with gentle shaking, followed by washing with TBS-T three times for 10 min each at RT. Images were visualized by an Odyssey imager (LI-COR) and processed using ImageStudio Lite v5.2 (LI-COR). Antibodies used in this study include: anti-ATF4 (ProteinTech, 28657-1-AP, rabbit, 1:1000), anti-β-actin (ProteinTech, 66009-1-Ig, mouse, 1:5000), anti-β-actin

(ProteinTech, 81115-1-RR, rabbit, 1:5000), anti-COX IV (Invitrogen, MA5-17279, mouse, 1:2000), anti-V5 (Thermo Fisher Scientific, R96025, mouse, 1:1000), anti-eIF2α (ProteinTech, 11170-1-AP, Rabbit, 1:1000), anti-GFP (Roche, 11814460001, mouse, 1:1,000), anti-HSPD1 (Invitrogen, MA3-012, mouse, 1:2000), anti-LC3 (ProteinTech, 14600-1-AP, Rabbit, 1:500), anti-phospho-eIF2α(ser51) (Cell Signaling Technology, 3398S, Rabbit, 1:1000), anti-Parkin (Invitrogen, 702785, rabbit, 1:500), anti-PINK1 (ProteinTech, 23274-1-AP, rabbit, 1:500), anti-PINK1 (CST, 6946S, rabbit, 1:1000), anti-SDHB (Invitrogen, MA5-26936, mouse, 1:1000), anti-TIMM23 (Proteintech, 67535-1-Ig, mouse, 1:1000), anti-MFN1 (Invitrogen, PA5-38042, rabbit, 1:1000), anti-MFN2 (Proteintech, 12186-1-AP, rabbit, 1:1000), anti-phospho-Ubiquitin (CST, 62802S, rabbit, 1:1000), anti-OMA1 (Proteintech, 17116-1-AP, rabbit, 1:1000), anti-HRI (Proteintech, 20499-1-AP, rabbit, 1:1000), anti-VDAC (Cell Signaling Technology, 4661S, Rabbit, 1:1000). Antibodies used in Supplementary Fig. 4 were: mouse monoclonal anti-EEA1 (BD Biosciences, Cat# 610456, 1:4000 dilution), mouse monoclonal anti-Kinectin (Santa Cruz Biotechnology, sc-374576, 1:5000), rabbit monoclonal anti-PERK (Cell Signaling Technology, 3192, 1 :2000), mouse monoclonal anti-GM130 (BD Biosciences, 610822, 1:2000), mouse monoclonal anti-LAMP2 (Santa Cruz Biotechnology, sc-18822, 1:5000), mouse monoclonal anti-calnexin (BD Biosciences, 610524, 1:4000 dilution), rabbit polyclonal anti-HRI (Proteintech, 20499-1-AP, 1 :2000), mouse monoclonal anti-calreticulin (BD Biosciences, 612137, 1:4000 dilution), rabbit monoclonal anti-ATF4 (Cell Signaling Technology, 11815, 1 :2000), rabbit monoclonal anti-Tubulin (Cell Signaling Technology, 2128, 1 :10000), rabbit polyclonal anti-DELE1 (Thermo Fisher Scientific, PA5-57712, 1 :1000), rabbit monoclonal anti-Syntaxin6 (Cell Signaling Technology, 2869, 1:5000), rabbit monoclonal anti-Phospho-eIF2a (Ser51) (Cell Signaling Technology, 3398, 1 :2000), rabbit monoclonal eIF2α (Cell Signaling Technology, 5324, 1 :5000), mouse monoclonal anti-GAPDH (Sigma-Aldrich, G8795, 1 :10000) and rabbit polyclonal anti-VDAC (Millipore, AB10527, 1:10000). All uncropped Western blot scans are included in the Source Data.

## Immunofluorescence

Cells were grown on coverslips. They were fixed by incubation in 4 % paraformaldehyde in PBS for 10 min, and then permeabilized by incubation with 0.15 % Triton X-100 in PBS for 15 min. Non-specific binding sites were blocked by incubating cells in a solution of 2 % BSA in PBS for 1 h. The cells were then incubated for 1 h at room temperature or overnight at 4 °C with the primary antibodies. They were washed three times, for five min each, in PBS and were then incubated for 45 min with the specific Alexa Fluor-conjugated secondary antibodies (Invitrogen, Life Technologies). Nuclei were stained with DAPI (Sigma) and cells were again washed three times with PBS. Images were acquired with a Leica SP5 confocal microscope (Leica Microsystems) equipped with a 63× oil immersion fluorescence objective. The primary antibodies used for immunofluorescence were rabbit monoclonal anti-Phospho-eIF2a (Ser51) (Cell Signaling Technology, Cat# 3398, 1 :400 dilution) and mouse monoclonal anti-Hsp60 (BD Biosciences, Cat# 611563, 1:400).

## Mitochondrial membrane potential measurement

Cells were seeded at 25% confluency in 24-well plates (Thermo-Scientific, 930186) 24 h before drug treatment. To depolarize mitochondrial membrane, cells were treated with 10 μM CCCP (Sigma-Aldrich, C2759) for 3 or 24 h as a positive control. For mitochondrial stress, cells were incubated with 1.25 ng/ml oligomycin, 100 nM rotenone or 100 nM antimycin A for 24 h. To measure mitochondrial membrane potential, cells were stained with 100 nM tetramethylrhodamine (TMRE) (Invitrogen, T669) in DMEM media for 20 minutes at 37 °C. After staining, cells were harvested by trypsinization and neutralized with DMEM (Gibco, 2910058) media, then washed twice with D-PBS (with 0.5 % FBS). Cells were flowed using Attune CytPix flow cytometer.

TMRE signal was detected through a 561 nm laser and 574 nm emission filter. Recorded data was analyzed using FlowJo software. Cells were gated by forward and side scatter for viable, single cells. BFP signals indicate the cells containing sgRNA for CRISPRi.

## Live cell imaging and quantification

Cells were seeded at 25% confluence on 8-chamber glass slides (Cellvis, C8-1.5H-N). For induction of MTS-YFP during mitochondrial stress, cells were incubated with 500 ng/mL doxycycline (Sigma-Aldrich, 631311) and 1.25 ng/ml oligomycin (Sigma-Aldrich, 75351) overnight. Cells were stained with 5 μg/mL Hoechst 33342 (Invitrogen, H1399) to staining nucleus for 0.5 h before imaging. Imaging was performed with a Zeiss 780 confocal microscope (Carl Zeiss) equipped with a 63× oil immersion objective. Images were obtained using the ZEN software (Carl Zeiss). Within every experiment, laser percentage and exposure time for each channel were set the same across all samples in comparison. All images were analyzed using Fiji. Around 300 cells were analyzed for each condition, and cells with obvious GFP diffusion are considered as compromised import.

## Mitochondria isolation and mitochondrial protein in vitro import assay

After the indicated treatments, the cells were washed with cold PBS twice and transferred into 1.5-ml tubes. The cell pellets were resuspended in cold Solution A (220 mM mannitol, 70 mM sucrose, 20 mM HEPES-KOH, 1 mM EDTA and 2 mg/ml BSA) containing protease inhibitor cocktail (Roche) and incubated on ice for 10 min and then homogenized with a KONTES glass grinder on ice. The homogenate was centrifuged at 800 g, 4 °C for 10 min to remove the debris and nucleus; the supernatant was centrifuged at 10,000 g, 4 °C for 20 min to obtain the crude mitochondrial pellets. The crude mitochondrial fractions were then washed with cold Solution B (220 mM mannitol, 70 mM sucrose and 20 mM HEPES-KOH) twice. Mitochondria were resuspended in import buffer (250 mM Sucrose, 80 mM KCl, 5 mM ATP, 10 mM sodium succinate, 20 mM HEPES-KOH, 5 mM MgCl$_2$ and 2 mM DTT) to measure the concentration. The MTS$^{Su9}$-DHFR-Flag-His protein was synthesized in vitro using the T7 promoter TNT Quick Coupled Transcription/Translation System (Promega) at 30 °C for 90 min. Isolated mitochondria were incubated with the fresh in vitro synthesized protein in import buffer at 30 °C for the indicated time periods. The membrane potential was dissipated by CCCP treatment (100 μM) for 5 min as negative control. Import reactions were stopped by 5 min 10,000 g centrifuge at 4 °C. For digesting the non-imported in vitro synthesized proteins, the mitochondria were treated with proteinase K (10 μg/ml) for 20 min at 4 °C. The proteinase K digestion was stopped by adding 1 mM PMSF. After import reaction or proteinase K digestion, mitochondria were resuspended in RIPA buffer with 1X LDS sample buffer, and proteins were analyzed by SDS–PAGE and immunoblotting.

## Cell viability assay

96-well plates were coated with 50 μg/ml poly-D-Lysine (Gibco, A3890401) for 1 h at RT. After washing with PBS for twice and sterile water for once, the plates were left to dry at RT for 2 h. HEK293T cells were seeded in the coated plates at a density of 2 × 10$^4$ cells/well. For drug treatment assay, the cells were cultured for 24 hrs before 1.25 ng/ml oligomycin was added. For glucose deprivation assay, DMEM without glucose medium (Gibco, 11966025) was added at the same time as the cell seeding. Cell viability was determined by the crystal violet staining method. Briefly, crystal violet powder (Fisher Scientific, C581-25) was dissolved in water to prepare a 0.1% crystal violet solution (stored in dark). Cells were fixed with 4% paraformaldehyde (Electron Microscopy Sciences, 15714-S) for 10 min at RT. After washing twice with PBS to remove paraformaldehyde, the cells were stained with 0.1% crystal violet solution for 20 min at RT. The excess stain was removed

by washing with PBS thoroughly, and the stained cells were solubilized in 95% ethanol (200 μL for each well). 20 μL of the solution from each well was transferred to a new 96-well plate and diluted 5-fold by adding 80 μL of 95% ethanol. Finally, the absorbance was measured at 562 nm using a spectrophotometer (Spark Multimode Microplate Reader). The images were obtained by a ChemiDoc Imaging System (Bio-Rad).

## RT-qPCR

Total RNA was extracted using the Quick-RNA Miniprep Kit (Zymo Research, R1055), and first-strand cDNA was synthesized with Maxima H Minus First Strand cDNA Synthesis Kit (Thermo Scientific™, K1652). qPCR was performed with Applied Biosystems™ PowerUp™ SYBR™ Green Master Mix for qPCR (Applied Biosystems™, A25742). Fold changes in expression were calculated using the ΔΔCt method. qPCR primers used are as following:

Actin_F: 5′-GTCATCACCATTGGCAATGAG-3′
Actin_R: 5′-CGTCATACTCCTGCTTGCTG-3′
ASNS_F: 5′-ATCACTGTCGGGATGTACCC-3′
ASNS_R: 5′-TGATAAAAGGCAGCCAATCC-3′
DDIT3_F: 5′-AGCCAAAATCAGAGCTGGAA-3′
DDIT3_R: 5′-TGGATCAGTCTGGAAAAGCA-3′
HRI_F: 5′-ACACCAACACATACGTCCAG-3′
HRI_R: 5′-GCTCCATTTCTGTTCCAAACG-3′
DELE1_F: 5′-AGGCTGTGACTTCCATTCAG-3′
DELE1_F: 5′-TCGCCACTCTTCATGTTCTC-3′

## Statistics and reproducibility

No statistical method was used to predetermine sample size. No data were excluded from the analyses. The experiments were not randomized. The Investigators were not blinded to allocation during experiments and outcome assessment. Statistics were performed on biological repeats via Two-way ANOVA test followed by Turkey's multiple comparisons test (two-sided) using GraphPad Prism version 10. All figures show representative data from experiments independently repeated at least once on different days.

## Reporting summary

Further information on research design is available in the Nature Portfolio Reporting Summary linked to this article.

# Data availability

All other data supporting the findings of this study are in the Supplementary Figs. Source data are provided with this paper. Source data include uncropped western blot scans, flow cytometry and qPCR data. Source data are provided with this paper.

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

## Acknowledgements

We thank Dr. Ruilin Tian, Dr. Emmy Li and Dr. Bernard Cook for critical reading of this manuscript. We thank Dr. Evan Jellison and Li Zhu from the UConn Health Flow Cytometry Facility for their technical support on cell sorting. We thank Dr. Yi Wu and Susan Staurovsky from the CCAM Microscopy Facility for their technical support on imaging. We thank Dr. Kazuya Machida for his expert assistance with protein purification intended for the cell-free import assays. This work was funded by the startup from UConn Health (XG), the Research Excellence Program from University of Connecticut (XG) and the National institutes of Health (R35GM155240 to XG), and Agence Nationale pour la Recherche (ANR-22-CE13-0032, MITOCISR to DA).

## Author contributions

M.Y., Z.M., D.A. and X.G. conceived, designed, performed, and interpreted the experiments. K.W., W.L., and I.N.I. performed experiments. X.G. supervised the project. M.Y., Z.M., D.A. and X.G. wrote the manuscript.

## Competing interests

We declare no conflicts related to the work described in this manuscript.
