## [Transparent Peer Review file · Nature Communications]

The Integrated Stress Response Suppresses PINK1-dependent Mitophagy by Preserving Mitochondrial Import Efficiency

Corresponding Author: Dr Xiaoyan Guo

Version 0:

Reviewer comments:

Reviewer #3

(Remarks to the Author)

The manuscript by Yang et al. reports on the integrated stress response (ISR) suppressing PINK1-dependent mitophagy during mild to moderate mitochondrial stress by limiting protein synthesis and preserving mitochondrial protein import. When ISR is absent, elevated protein synthesis overwhelms import machinery, causing PINK1 accumulation and triggering mitophagy, whereas severe stress impairs import regardless of ISR.

Major:

1. Figure 3i shows an in vitro mitochondrial import assay. This is the key assay to support the claims. Surprisingly, it appears that only oligomycin-treated mitochondria were analyzed? Where is the essential mock-treated control? Without it, it is impossible to determine if this is indeed a mitochondrial stress phenotype or merely a baseline shift in DELE1 KD cells.
2. Along similar lines, DELE1 KD cells appear to import less COX8-YFP in Figure 3C, indicating this may be a general phenotype instead of a DELE1-dependent stress induced import block.
3. A HSPD1 and PINK1 control western blot for Figure 4p is missing to demonstrate that the overexpressed mitochondrial precursors do indeed trigger/increase mitophagy by blocking the import of other mitochondrial proteins. If this were the case, why is overexpression of these precursors not sufficient to increase mitophagy in control cells treated with low doses of oligomycin?
4. It appears that the data represented in numerous figure panels show technical repeats instead of biological replicates ("mean +/- SD from three culture wells"), yet statistical calculations are performed. It is not standard practice (and somewhat meaningless) to perform statistical calculations on technical repeats. While it is understandable to show representative repeats for non-quantitative data such as Western blots, it is also not standard practice to show representative repeats for quantitative data such as flow cytometry measurements. Biological replicates are needed to support statistical assessment. This concerns Main Figures 1F, 2A, 2E, 3E, 3H, 4A, 4C, 4E, 4G, 4K, 4i, 5A, 6C, 6D, and numerous panels in the Extended Data.
5. In the current form, the manuscript lacks key mechanistic insight into how the ISR might maintain protein import (at the TOM complex or further upstream or downstream) in the investigated settings. Of note, other studies have previously reported that ISR inhibition during stress can lead to TOM clogging.

Minor:

1. "DFP induces only mild mitophagy, which is slightly enhanced by DELE1 KD in a PRKN-dosage-dependent manner" This should read "Parkin dosage-independent" as no difference between Parkin low and high cells is visible, which is consistent with previous studies.

Reviewer #4

(Remarks to the Author)

1. We believe that reviewer1/2's concerns regarding the experimental design and analysis strategy related to the split-GFP

experiments are justified. In the latest manuscript, the authors introduced the MTSdiablo-GFP11 system, and the data conclusions obtained from this system can well address the impact of oligomycin on mitochondrial protein input. Furthermore, Extended Data Fig. 6a also provides a satisfactory answer to the reproducibility of the Oligomycin condition, although the effect is not as good as that of CCCP. We must admit that the Oligomycin system is not an ideal experimental design for studying this scientific issue. The authors should consider transferring the relevant data of Oligomycin to Extended Data to avoid causing comprehension difficulties for readers when they read the article. Therefore, the newly added data is now convincing.

2. In the "Reviewer 3" section, in terms of functional exports, whether it is Parkinson's disease or cardiomyopathy, if the research on disease models can be introduced and the universality of the new concepts can be demonstrated, it should be more persuasive.

REVIEWER COMMENTS

Reviewer #3 (Remarks to the Author):

The manuscript by Yang et al. reports on the integrated stress response (ISR) suppressing PINK1-dependent mitophagy during mild to moderate mitochondrial stress by limiting protein synthesis and preserving mitochondrial protein import. When ISR is absent, elevated protein synthesis overwhelms import machinery, causing PINK1 accumulation and triggering mitophagy, whereas severe stress impairs import regardless of ISR.

Major:

1. Figure 3i shows an in vitro mitochondrial import assay. This is the key assay to support the claims. Surprisingly, it appears that only oligomycin-treated mitochondria were analyzed? Where is the essential mock-treated control? Without it, it is impossible to determine if this is indeed a mitochondrial stress phenotype or merely a baseline shift in DELE1 KD cells.

We greatly appreciate the reviewer's comments and careful re-evaluation of our work. We agree that it is critical to address DELE1's role on mitochondrial protein import under physiological conditions. To address this, we have now included a new cell-free

mitochondrial protein import assay using mitochondria from both WT and *DELE1* KD cells, either left untreated or treated with oligomycin for 24hr. From this new experiment, we were able to reproduce results consistent with our previous observations: mitochondria from oligomycin-treated *DELE1* KD cells treated with oligomycin are significantly compromised in protein import.

More importantly, **we now show DELE1 does not affect mitochondrial protein import under physiological condition.**

Interestingly, mitochondria from *DELE1* KD cells have slightly better import, possibly due to the

elimination of DELE1 as a substrate that could otherwise occupy the mitochondrial translocases.

The new experiments were repeated twice and are included in the Extended Data Fig. 8a. The corresponding text is highlighted in the Results section, “**DELE1 maintains mitochondrial protein import efficiency under mild stress conditions**”.

2. Along similar lines, DELE1 KD cells appear to import less COX8-YFP in Figure 3C, indicating this may be a general phenotype instead of a DELE1-dependent stress induced import block.

As shown by our new cell-free import assay discussed in our previous responses, DELE1 is not essential to maintain mitochondrial protein import under non-stressed condition. In Fig. 3c, we quantified the percentage of cells exhibiting diffuse fluorescence, rather than fluorescence intensity. As noted by the reviewer, the reduced MTS^{COX8}-YFP signal in *DELE1* KD cells without any stress likely reflects technical factors such as photobleaching. Importantly, no cells showed a diffuse cytosolic distribution of the fluorescent reporter, indicating that mitochondrial import remained efficient. To avoid confusion for readers, we manually adjusted the brightness of this image in Fiji to a comparable level with other images.

3. A HSPD1 and PINK1 control western blot for Figure 4p is missing to demonstrate that the overexpressed mitochondrial precursors do indeed trigger/increase mitophagy by blocking the import of other mitochondrial proteins. If this were the case, why is overexpression of these precursors not sufficient to increase mitophagy in control cells treated with low doses of oligomycin?

Extended Data Fig.13 addresses this question.

Over expression of TIMM44 significantly increases mitophagy under untreated condition, however, mitophagy in these cells can be further promoted by ablation of the ISR. This is because overexpression of TIMM44 alone is sufficient to trigger the ISR even in the absence of additional pharmacological stress, as indicated by upregulation of ATF4 (lane 5). Consequently, global translation is attenuated, which improves import and suppress mitophagy.

4. It appears that the data represented in numerous figure panels show technical repeats instead of biological replicates (“mean +/- SD from three culture wells”), yet statistical calculations are performed. It is not standard practice (and somewhat meaningless) to perform statistical calculations on technical repeats. While it is understandable to show representative repeats for non-quantitative data such as Western blots, it is also not standard practice to show representative repeats for

quantitative data such flow cytometry measurements. Biological replicates are needed to support statistical assessment. This concerns Main Figures 1F, 2A, 2E, 3E, 3H, 4A, 4C, 4E, 4G, 4K, 4i, 5A, 6C, 6D, and numerous panels in the Extended Data.

We thank the reviewer for raising this important point and agree that statistical analyses should be based on independent biological replicates. We apologize for the lack of clarity in the original manuscript regarding how replicates were defined.

Our approach follows the presentation of flow cytometry data in previous high-impact studies on mitophagy (Michaelis, J.B., et al., *Nat Commun* 13, 5164, 2022, PMID: 36056001; Ham, S.J., et al., *Mol Cell* 85, 2287-2302, 2025, PMID: 40505663). In our experiments, cells were seeded into separate culture wells and treated independently, with each well undergoing independent drug treatment, processing, and measurement (including staining, acquisition, and flow cytometry analysis). Each well therefore represents an independently handled cell population and was treated as an independent experimental unit.

We recognize that the term “biological replicate” can be interpreted differently and have now clarified this in all figure legends and methods. Specifically, we distinguish between independently treated culture wells and experiments performed on separate days. All these figures present representative data from experiments independently repeated at least once on different days.

5. In the current form, the manuscript lacks key mechanistic insight into how the ISR might maintain protein import (at the TOM complex or further upstream or downstream) in the investigated settings. Of note, other studies have previously reported that ISR inhibition during stress can lead to TOM clogging.

We thank the reviewer for this point. Our study provides a detailed mechanism by which the integrated stress response (ISR) maintains mitochondrial protein import during mitochondrial stress through attenuation of protein synthesis. Using genetic ablation and pathway analyses, we provide strong evidence that ISR-mediated preservation of protein import occurs independently of ATF4 upregulation. Through pharmacological manipulation of protein synthesis, we demonstrate for the first time that translational attenuation during mitochondrial stress directly contributes to maintaining protein import.

Based on previous studies showing that artificial overexpression of certain mitochondrial precursor proteins can clog protein translocases, we propose that ISR-induced attenuation of protein synthesis during mitochondrial stress reduces the trafficking burden through the TOM complex. Importantly, prior studies have not definitely

suggested or prove that ISR inhibition itself causes TOM clogging or compromise protein import; rather, ISR inhibition increases the synthesis of mitochondrial precursors, thereby facilitating the detection of import failure.

Furthermore, our study links this mechanism to mitophagy, revealing how distinct mitochondrial quality control pathways are coordinated to regulate cellular stress responses.

Minor:

1. “DFP induces only mild mitophagy, which is slightly enhanced by DELE1 KD in a PRKN-dosage-dependent manner” This should read “Parkin dosage-independent” as no difference between Parkin low and high cells is visible, which is consistent with previous studies.

We thank the reviewer for identifying this issue and have corrected it in the revised manuscript.

Reviewer #4 (Remarks to the Author):

1. We believe that reviewer1/2's concerns regarding the experimental design and analysis strategy related to the split-GFP experiments are justified. In the latest manuscript, the authors introduced the MTSdiablo-GFP11 system, and the data conclusions obtained from this system can well address the impact of oligomycin on mitochondrial protein input. Furthermore, Extended Data Fig. 6a also provides a satisfactory answer to the reproducibility of the Oligomycin condition, although the effect is not as good as that of CCCP. We must admit that the Oligomycin system is not an ideal experimental design for studying this scientific issue. The authors should consider transferring the relevant data of Oligomycin to Extended Data to avoid causing comprehension difficulties for readers when they read the article. Therefore, the newly added data is now convincing.

We thank the reviewer for their comment on our experimental design. We employed two versions of the split-GFP protein import reporters (one with GFP1-10 localized to cytosol and the other with GFP1-10 localized to mitochondria) to address previous reviewers' concern regarding the measurement of mitochondrial protein import. In addition, we have incorporated cell-free import assays to further validated our conclusions.

We respectfully disagree with the comment that oligomycin system is not an ideal experiment design in our study. In contrast to CCCP, oligomycin and other ETC inhibitors induce mild mitochondrial stress, which allows us to specifically examine both

the negative regulatory role of the ISR in mitophagy, another key mitochondrial quality control pathway, and its positive role in maintaining protein import.

2. In the "Reviewer 3" section, in terms of functional exports, whether it is Parkinson's disease or cardiomyopathy, if the research on disease models can be introduced and the universality of the new concepts can be demonstrated, it should be more persuasive.

We thank the reviewer for this thoughtful suggestion. We agree that disease models can provide important functional context. In the current manuscript, we have already discussed the relevance of our findings to disease settings - ***In several mouse models of mitochondrial cardiomyopathy, outer mitochondrial membrane rupture and mitophagy were exclusively observed in the absence of DELE1⁶⁷. Future studies are needed to determine whether mitophagy regulation in these mouse models contributes to the protective role of the DELE1 pathway in cardiomyopathy.*** Our goal in this study was to establish the underlying molecular mechanism, which we believe is broadly applicable given the conserved nature of the pathway. We therefore focused on mechanistic dissection rather than disease-specific models, while outlining the implications and future directions in relevant disease contexts.